# A process-based evaluation of biases in extratropical stratosphere–troposphere coupling in subseasonal forecast systems

**Chaim I. Garfinkel**[1], **Zachary D. Lawrence**[2,3], **Amy H. Butler**[4], **Etienne Dunn-Sigouin**[5,6], **Irina Statnaia**[7], **Alexey Y. Karpechko**[7], **Gerbrand Koren**[8], **Marta Abalos**[9], **Blanca Ayarzagüena**[9], **David Barriopedro**[10], **Natalia Calvo**[9], **Alvaro de la Cámara**[9], **Andrew Charlton-Perez**[11], **Judah Cohen**[12], **Daniela I. V. Domeisen**[13,14], **Javier García-Serrano**[15], **Neil P. Hindley**[16], **Martin Jucker**[17], **Hera Kim**[18], **Robert W. Lee**[11], **Simon H. Lee**[19], **Marisol Osman**[20,21], **Froila M. Palmeiro**[15], **Inna Polichtchouk**[22], **Jian Rao**[23], **Jadwiga H. Richter**[24], **Chen Schwartz**[3], **Seok-Woo Son**[18], **Masakazu Taguchi**[25], **Nicholas L. Tyrrell**[7], **Corwin J. Wright**[16], and **Rachel W.-Y. Wu**[14]

[1]Fredy & Nadine Herrmann Institute of Earth Sciences, The Hebrew University of Jerusalem, Jerusalem, Israel
[2]Cooperative Institute for Research in Environmental Sciences (CIRES), University of Colorado Boulder, Boulder, CO, USA
[3]NOAA Physical Sciences Laboratory (PSL), Boulder, CO, USA
[4]NOAA Chemical Sciences Laboratory (CSL), Boulder, CO, USA
[5]NORCE Norwegian Research Centre, Bergen, Norway
[6]Bjerknes Centre for Climate Research, Bergen, Norway TS1
[7]Finnish Meteorological Institute, Meteorological Research, Helsinki, Finland
[8]Copernicus Institute of Sustainable Development, Utrecht University, Utrecht, the Netherlands
[9]Department of Earth Physics and Astrophysics, Universidad Complutense de Madrid, Madrid, Spain
[10]Instituto de Geociencias (IGEO), Consejo Superior de Investigaciones Científicas–Universidad Complutense de Madrid (CSIC-UCM), Madrid, Spain
[11]Department of Meteorology, University of Reading, Reading, UK
[12]Atmospheric and Environmental Research Inc., Lexington, MA, USA
[13]Faculty of Geosciences and Environment, University of Lausanne, Lausanne, Switzerland
[14]Institute for Atmospheric and Climate Science, ETH Zurich, Zurich, Switzerland
[15]Group of Meteorology, Universitat de Barcelona (UB), Barcelona, Spain
[16]Centre for Climate Adaptation & Environment Research, University of Bath, Bath, UK
[17]Climate Change Research Centre, ARC Centre of Excellence for Climate Extremes, University of New South Wales, Sydney, Australia
[18]School of Earth and Environmental Sciences, Seoul National University, Seoul, South Korea
[19]Department of Applied Physics and Applied Mathematics, Columbia University, New York, NY, USA
[20]CONICET–Universidad de Buenos Aires, Centro de Investigaciones del Mar y la Atmósfera (CIMA), Buenos Aires, Argentina
[21]CNRS–IRD–CONICET–UBA, Instituto Franco-Argentino de Estudios sobre el Clima y sus Impactos (IRL 3351, IFAECI), Buenos Aires, Argentina
[22]European Centre for Medium-Range Weather Forecasts, Reading, UK
[23]Collaborative Innovation Center on Forecast and Evaluation of Meteorological Disasters/Key Laboratory of Meteorological Disaster, CE1 Ministry of Education, Nanjing University of Information Science & Technology, Nanjing, China
[24]Climate and Global Dynamics Laboratory, National Center for Atmospheric Research, Boulder, CO, USA
[25]Department of Earth Science, Aichi University of Education, Kariya, Japan

**Correspondence:** Chaim I. Garfinkel (chaim.garfinkel@mail.huji.ac.il)

Received: 10 June 2024 – Discussion started: 27 June 2024
Revised: 30 October 2024 – Accepted: 12 November 2024 – Published:

**Abstract.** Two-way coupling between the stratosphere and troposphere is recognized as an important source of subseasonal-to-seasonal (S2S) predictability and can open windows of opportunity for improved forecasts. Model biases can, however, lead to a poor representation of such coupling processes; drifts in a model's circulation related to model biases, resolution, and parameterizations have the potential to feed back on the circulation and affect stratosphere–troposphere coupling. We introduce a set of diagnostics using readily available data that can be used to reveal these biases and then apply these diagnostics to 22 S2S forecast systems.

In the Northern Hemisphere, nearly all S2S forecast systems underestimate the strength of the observed upward coupling from the troposphere to the stratosphere, downward coupling within the stratosphere, and the persistence of lower-stratospheric temperature anomalies. While downward coupling from the lower stratosphere to the near surface is well represented in the multi-model ensemble mean, there is substantial intermodel spread likely related to how well each model represents tropospheric stationary waves.

In the Southern Hemisphere, the stratospheric vortex is oversensitive to upward-propagating wave flux in the forecast systems. Forecast systems generally overestimate the strength of downward coupling from the lower stratosphere to the troposphere, even as most underestimate the radiative persistence in the lower stratosphere. In both hemispheres, models with higher lids and a better representation of tropospheric quasi-stationary waves generally perform better at simulating these coupling processes.

# 1 Introduction

The extratropical stratosphere and troposphere are coupled through dynamical interactions between planetary-scale atmospheric Rossby waves and the mean flow. This vertical coupling operates in both directions: upward coupling from tropospheric variability induces variability in the stratosphere, while downward coupling from stratospheric variability can impact weather in the troposphere (Butler et al., 2019; Scaife et al., 2022). Both weak and strong polar stratospheric vortex extremes have been shown to influence surface climate and weather extremes for weeks to months afterwards (Domeisen and Butler, 2020) due to the long radiative timescales in the lower stratosphere (Hitchcock et al., 2013), which means that stratospheric variability can potentially provide windows of opportunity for prediction on subseasonal-to-seasonal (S2S) timescales (Butler et al., 2019; Domeisen et al., 2020b). However, model biases in either the troposphere or the stratosphere can impact these coupling processes, compromising opportunities to increase S2S predictability that could otherwise be achieved. The goal of this study is to identify systematic biases in extratropi-

cal stratosphere–troposphere coupling processes across S2S forecast systems.

Variability in the upward flux of planetary-scale (wavenumbers 1–3) Rossby waves drives variability in the stratospheric polar vortex. Upward wave propagation is strengthened when the wave (or eddy) constructively interferes with the climatological stationary wave pattern, while weakened wave flux occurs when the linear interference is destructive (Garfinkel et al., 2010; Smith and Kushner, 2012). In addition, Rossby waves can amplify or weaken due to nonlinear processes (Scinocca and Haynes, 1998; Boljka and Birner, 2020). Rossby waves can only propagate upward into the stratosphere when the zonal flow is westerly but below a critical wind speed (Charney and Drazin, 1961), conditions that occur primarily in Northern Hemisphere (NH) extended winter (November–March) and Southern Hemisphere (SH) spring (September–November). A weaker upward flux of wave activity can lead to a strengthening of the polar vortex (Limpasuvan et al., 2004). On the other hand, an anomalously strong or persistent pulse of wave activity can weaken, and even reverse, the westerly winds of the vortex (Andrews et al., 1987; Polvani and Waugh, 2004; Garfinkel et al., 2010). In particular, about once every 2 years the Arctic polar vortex completely breaks down and the zonal winds reverse direction in an extreme event called sudden stratospheric warming (SSW) (Baldwin et al., 2021). In SH spring, this upward coupling more typically manifests as a modulation of the timing of the seasonal polar vortex breakdown, with weaker upward flux of wave activity resulting in a delayed breakdown in spring and vice versa for stronger upward wave flux (Byrne and Shepherd, 2018; Lim et al., 2018). A complete breakdown of the SH vortex has only been observed once, in September 2002.

Variability in the strength and location of the stratospheric polar vortex can also exert a downward influence on weather patterns (Boville, 1984; Haynes et al., 1991; Hitchcock and Simpson, 2014). Near the tropopause, interactions of the stratospheric signal with both transient and stationary eddies are important for communicating the signal to the surface (Song and Robinson, 2004; Domeisen et al., 2013; White et al., 2020, 2022). While both stratospheric and tropospheric factors influence the downward communication of the signal (Afargan-Gerstman et al., 2022), the exact mechanism of downward coupling remains unclear.

Accurately simulating both upward and downward vertical coupling requires reasonably accurate simulation of processes such as the location and strength of stationary planetary waves and the jet in the troposphere (Schwartz et al., 2022), the strength and seasonality of stratospheric wind speeds, and the radiative timescales of the lower stratosphere. Recently, Lawrence et al. (2022) identified systematic stratospheric biases across S2S forecast systems. In particular, they found that most forecast systems exhibit a warm bias in the

global-mean stratosphere and a cold bias in the extratropical lower stratosphere–upper troposphere. These biases were suggested to be due to biases in radiative heating rates associated with model biases in ozone and water vapor (cf. Bland et al., 2021). Most forecast systems also showed strong and cold polar vortex biases, which suggests that there are underlying difficulties in accurately representing vertical coupling processes. In general, stratospheric biases were substantially worse for models with a low model lid height, a long-standing issue (Lawrence, 1997; Marshall and Scaife, 2010) that has also been identified in seasonal prediction systems (Butler et al., 2016) and climate models (Charlton-Perez et al., 2013), which can be exacerbated by poorly designed physics parameterizations (Shaw and Perlwitz, 2010).

While systematic biases in the stratosphere were detailed in Lawrence et al. (2022), a deeper exploration of how S2S models simulate the processes that underlie stratosphere–troposphere vertical coupling is warranted, given that these processes ultimately drive the impacts on surface weather patterns and regional hazards. As part of the collaborative effort of the World Climate Research Programme (WCRP) Atmospheric Processes And their Role in Climate (APARC) Stratospheric Network for the Assessment of Predictability (SNAP) project, we investigate how extratropical atmospheric biases are linked to the simulation of stratosphere–troposphere coupling in S2S forecast systems. After introducing the data and methods in Sect. 2, we demonstrate that many S2S forecast systems struggle to represent the strength of observed upward coupling from the troposphere to the stratosphere (Sect. 3.1), the sensitivity of the stratospheric polar vortex to upward wave flux (Sect. 3.2), interannual variability in heat flux extremes (Sect. 3.3), downward coupling within the stratosphere (Sect. 3.4), and downward coupling from the lower stratosphere to the surface (Sect. 3.5). After considering possible factors that can account for the inter-model spread in coupling strength (Sect. 3.6), we summarize our results and place them in the context of previous work (Sect. 4).

## 2 Data and methods

### 2.1 Subseasonal-to-seasonal (S2S) hindcast and reanalysis datasets

We use ensemble hindcast data from the S2S Prediction Project Database (Vitart et al., 2017) and, depending on data availability, select forecast systems not included in the S2S database: (i) the National Oceanic and Atmospheric Administration's Global Ensemble Forecast System version 12 (NOAA GEFSv12; Hamill et al., 2021; Guan et al., 2022), (ii) the National Center for Atmospheric Research Community Earth System Model version 2 (CESM2) with version 6 of the Community Atmosphere Model as its atmospheric component (NCAR CESM2–CAM6, hereafter

CESM2–CAM), and (iii) CESM2 with version 6 of the Whole Atmosphere Community Climate Model as its atmospheric component (CESM2–WACCM6, hereafter CESM2–WACCM; Richter et al., 2022). Daily gridded latitude–longitude data were only retained for the seven forecast systems that provide at least 35 d forecasts to the S2S database due to the large data volume, and so metrics which rely on these data are only computed for these seven systems.

Lawrence et al. (2022) analyzed biases over the period common to all models (1999–2010), but here we include upgraded versions of several models, for which the hindcasts begin several years after 1999. Furthermore, the specific days on which forecasts are initialized differ across systems even for a given year. We therefore have elected not to focus on a common period in this paper except for the analyses in Sect. 3.3. The specific model versions and the period used for each system are included in Table 1, and their vertical resolutions are detailed in Fig. 1. For "pixel figures" quantifying biases in individual systems (e.g., Figs. 2, 4), we subsample reanalysis data to match each system, thus allowing us to pinpoint biases. For figures showing lagged correlations and lagged regression, we show the mean across the forecasting systems of the subsampled coupling strength with a solid black line and the spread in the subsampled coupling strength across the available S2S systems with a vertical thin line; because there is no exact overlap in the analysis period, model biases should not be inferred from face value from these lagged correlation/regression figures. Nonetheless, these thin vertical lines offer an estimate of the range of sampling variability in ERA5, and thus if a given model lies outside of this range, a bias can be even more confidently detected as there is no longer another reasonable explanation.

The subseasonal hindcasts analyzed here are initialized with different atmospheric datasets. To ensure this has no significant effects on our results, we compare the hindcast fields to those from the ERA5 reanalysis (Hersbach et al., 2020) so that comparisons and biases are all determined with respect to a consistent dataset. Note that for the time periods and levels considered here (post-1990 and up to 10 hPa) most modern reanalysis products are in good agreement (Long et al., 2017; Gerber and Martineau, 2018; Ayarzagüena et al., 2019; Fujiwara, M. et al., 2021), and thus our results should be robust across reanalyses.

### 2.2 Methods

We use the following eight key metrics to diagnose coupling strength throughout this paper: $\overline{v_{k=1}T_{k=1}}$ at 500 and 100 hPa; $\overline{v_{k=2}T_{k=2}}$ at 500 and 100 hPa; polar-cap height (60° pole, hereafter Zcap) at 10, 100, and 850 hPa; and polar-cap temperature (60° pole, hereafter Tcap) at 100 hPa. $v$ denotes the meridional wind, $T$ is the temperature, and $\bar{}$ is the zonal mean. We decompose $v$ and $T$ by wavenumber before computing their product, e.g., $\overline{v_{k=1}T_{k=1}}$ for zonal wavenumber 1 (wave 1) heat flux. The upward flux of planetary waves is di-

**Table 1.** Details of the subseasonal-to-seasonal forecast systems used herein.

| Model | S2S database version(s) | Hindcast period | Initializations per season | Ensemble size | Forecast span | Model top |
|---|---|---|---|---|---|---|
| BoM* | POAMA P24 | 1999–2010 | 15 | 33 | 62 d | 10 hPa |
| CESM2–CAM* | – | 1999–2019 | 12–13 | 11 | 45 d | 2 hPa |
| CESM2–WACCM | – | 1999–2019 | 12–13 | 5 | 45 d | $4.5 \times 10^{-6}$ hPa |
| CMA* | BCC-CPS-S2Sv1 | 1994–2014 | 90–91 | 4 | 60 d | 0.5 hPa |
| CMA | BCC-CPS-S2Sv2 | 2005–2019 | 25 | 4 | 60 d | 0.1 hPa |
| ISAC-CNR* | GLOBO | 1990–2010 | 18 | 5 | 32 d | 6.8 hPa |
| CNRM | CNRM-CM6-0 | 1993–2014 | 12 | 15 | 50 d | 0.01 hPa |
| CNRM | CNRM-CM6-1 | 1992–2017 | 13 | 10 | 47 d | 0.01 hPa |
| ECCC-lo* | GEPS 4 | 1995–2014 | 25–26 | 4 | 32 d | 2 hPa |
| ECCC-hi | GEPS 6 | 1998–2017 | 23–25 | 4 | 32 d | 0.1 hPa |
| ECCC-hi | GEPS 7 | 2001–2020 | 13–21 | 4 | 32 d | 0.1 hPa |
| ECMWF | CY45R1 | 1998–2018 | 26 | 11 | 46 d | 0.01 hPa |
| ECMWF | CY47R3 | 2002–2020 | 26 | 11 | 46 d | 0.01 hPa |
| GEFSv12 | – | 2000–2019 | 12–13 | 11 | 35 d | 0.1 hPa |
| HMCR | RUMS | 1991–2014 | 12–13 | 11 | 46 d | 0.04 hPa |
| JMA | GEPS1701 | 1990–2012 | 9 | 5 | 34 d | 0.01 hPa |
| JMA | CPS3 | 1991–2020 | 6 | 5 | 34 d | 0.01 hPa |
| KMA | GloSea5-GC2 | 1991–2016 | 12 | 3 | 60 d | 85 km |
| KMA | GloSea6-GC32 | 1993–2016 | 12 | 3 | 60 d | 85 km |
| NCEP | CFSv2 | 1999–2010 | 90–91 | 4 | 44 d | 0.02 hPa |
| UKMO | GloSea5 | 1993–2016 | 12 | 7 | 60 d | 85 km |
| UKMO | GloSea6 | 1993–2016 | 12 | 7 | 60 d | 85 km |

* Systems with low-top models. Note that we use the high-top HMCR RUMS model version with 96 vertical levels.

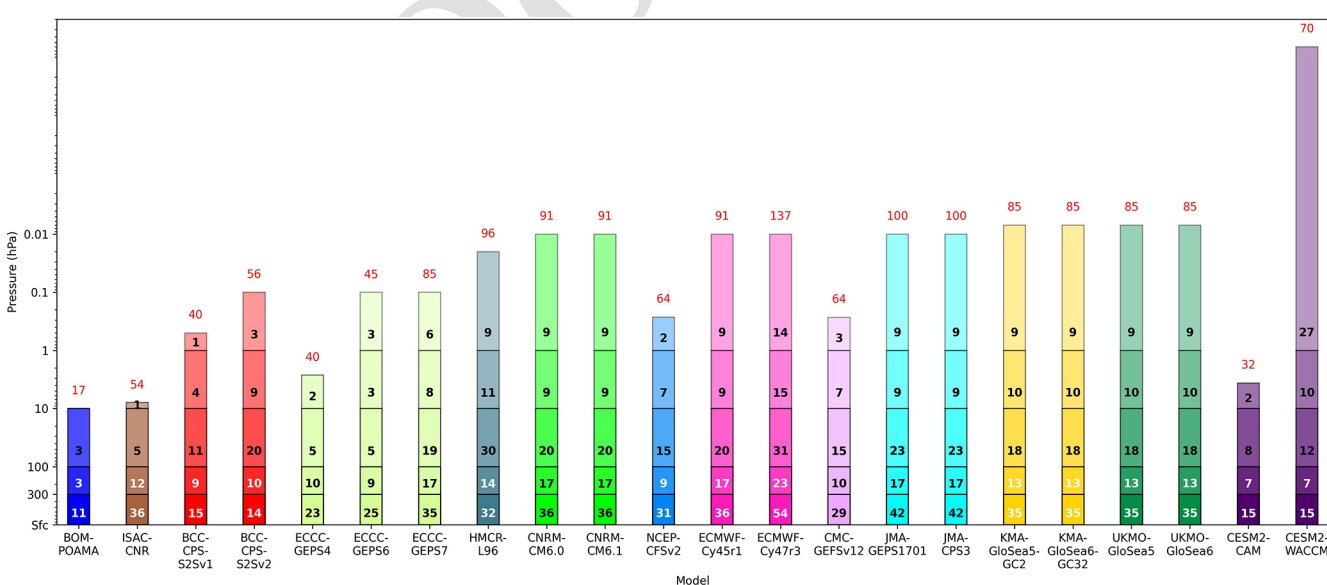

**Figure 1.** Schematic representation of model vertical resolution for all S2S prediction systems used in this study. Each block represents the pressure range indicated on the *y* axis. The number of model levels in each range is shown numerically (the font color was chosen for visualization and does not have additional meaning). The red number at the top of each bar shows the total number of levels in each model setup.

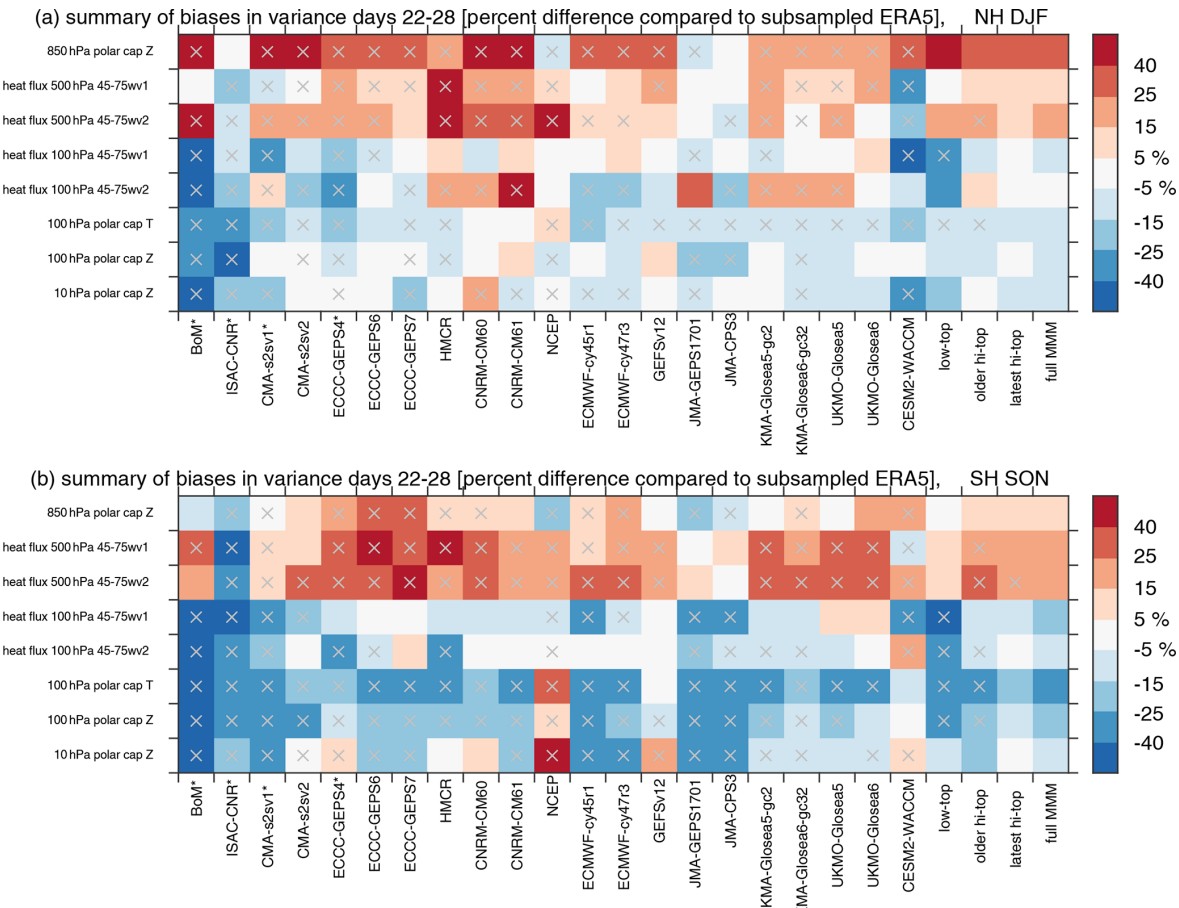

**Figure 2.** Variance of daily values of the various diagnostics in days 22–28 in the **(a)** Northern Hemisphere and **(b)** Southern Hemisphere. For each forecast system we compare the variance to that for the corresponding period in ERA5 and then show the percent error. A gray ×indicates models and metrics for which all ensemble members simulate variance that is either too weak or too strong or, alternatively, if the ERA5 variance does not fall within the envelope of the available members. The bias is defined as the difference in variance between the model and ERA5 divided by the variance in ERA5. The last four columns show the mean bias for low-top models, the older versions of high-top models, the latest versions of high-top models, and all models (full MMM); a gray × indicates all models agree on the sign of the bias. DJF: December–January–February. SON: September–October–November.

agnosed using the meridional eddy heat flux (e.g., $\overline{v_{k=1}T_{k=1}}$) rather than the vertical component of the Eliassen–Palm flux due to the limited vertical resolution available in the S2S archive.

S2S models typically archive data at coarse resolution due to the huge data volume. For models in the S2S database, we consider instantaneous daily values at 00:00 UTC on a $1.5° × 1.5°$ latitude–longitude grid, with 10 pressure levels between 1000 and 10 hPa. GEFSv12 data are provided 6-hourly on a $0.5° × 0.5°$ grid, with 25 pressure levels between 1000 and 1 hPa. CESM2–CAM and CESM2–WACCM provide zonally averaged daily fields at 192 latitudes ($\sim 0.9424°$ resolution) on the pressure levels closest to the model levels, which we interpolate to a set of 32 standard pressure levels between 1000 and 10 hPa. Heat flux data are not available for CESM2–CAM, and hence we show this model for limited diagnostics only. The eight diagnostics are computed on the available model grid.

As in Lawrence et al. (2022), we define forecast systems with model tops at or above 0.1 hPa, with several levels above 1 hPa as "high-top" models and all others as "low-top" models. Using this definition results in 17 forecast systems with high-top models and 5 forecast systems with low-top models (see Table 1); however, not all models are included for each analysis. Low-top models are identified with asterisks and/or dotted lines in the figures. We stress that the computation of high-top and low-top means is obtained from an unbalanced distribution of high-top and low-top models.

For each variable and forecast system, we derive lead-time-dependent climatologies, which we subtract from the raw forecast quantities to determine forecast anomalies. These climatologies are calculated by averaging all ensemble-mean hindcasts for a given day of a year and for

each lead time. For those systems providing a fixed set of hindcast initializations that do not uniformly cover the same days of a year in the hindcasts (e.g., GEFSv12 and CESM2), we permit differences of up to 3 d when creating the lead-time-dependent climatologies.

We quantify the tightness of coupling using both regression and correlation analyses. Regression coefficients directly diagnose the strength of coupling and are the closest we can get to answering questions such as "what is the heat flux anomaly at 100 hPa for a given heat flux anomaly at 500 hPa?". The downside of regression is that it is not possible to meaningfully compare the different coupling metrics in the paper to see their relative importance because the units are different. On the other hand, correlations normalize the units and allow for comparison between different metrics. Correlations also quantify how much of the linear variability between two quantities is shared. For most models and metrics, regression and correlation coefficients are similar. However, there are notable exceptions if a given model fails to simulate a reasonable amount of variance for a given metric. In these cases, we elect to use regression to diagnose coupling strength, as the correlation conflates two possible sources of error: error in the coupling strength with error in the underlying variance (the Results section provides several examples of such behavior). For completeness, in the Supplement we include figures that diagnose coupling strength using correlation. In all cases we calculate the regression and correlation coefficients for individual ensemble members first and then average over members.

Some models suffer from large ($> 40\,\%$) biases in variance, and so this concern about variance biases complicating the interpretation of correlations is difficult to sidestep. We demonstrate this in Fig. 2, which shows the percentage error in the daily variance in each forecast system for our eight key metrics and days 22–28 of the forecast. We compare each forecast system to the corresponding period in ERA5, and if all available ensemble members show a bias of the same sign, we indicate that pixel with a $\times$ symbol. Applying an $F$ test leads to a larger proportion of the pixels indicating significant biases (not shown). Most models overestimate variance in lower-tropospheric polar-cap height and tropospheric planetary wave heat flux in both hemispheres. In contrast, most underestimate variance in the lower stratosphere in the SH, and in the NH there is a notable decrease in the magnitude of the bias in variance from the troposphere up to the stratosphere. Most models also suffer from too little variance in lower-stratospheric polar-cap height and temperature in the SH. These biases in variance are qualitatively similar though weaker earlier in the integration (e.g., days 8–14, not shown). We are not aware of previous work that has found too strong a variance bias in the troposphere, and the causes and implications of these biases should be explored in future work.

## 3 Results

We now consider the relative abilities of the forecast systems in capturing the physical processes underlying stratosphere–troposphere coupling. To do this, we subdivide stratosphere–troposphere coupling into several components as follows and consider each individually below:

1. vertical propagation of planetary waves from the troposphere into the stratosphere

2. the sensitivity of the stratospheric polar vortex to upward wave driving from the lowermost stratosphere

3. sufficient interannual spread in daily heat flux extremes

4. downward propagation of stratospheric polar vortex anomalies from the upper and middle stratosphere to the lower stratosphere

5. the persistence of the polar vortex signal in the lower stratosphere that arises due to the long radiative timescales

6. downward propagation from the lower stratosphere to the near surface.

### 3.1 Vertical propagation of planetary waves from the troposphere into the stratosphere

We begin by considering the upward coupling of wave activity from the troposphere to the stratosphere. This coupling is quantified by computing the lagged correlation and regression coefficients between 500 and 100 hPa heat fluxes averaged over 45–75° in each hemisphere (Fig. 3a, b). The dominant direction of coupling is for tropospheric (500 hPa) heat fluxes to precede lower-stratospheric (100 hPa) heat fluxes.

In the NH, this coupling peaks when tropospheric heat flux precedes lower-stratospheric heat flux by 3 d for wave 1 CE3 and by 2 d for wave 2 in ERA5 (thick black lines in Fig. 3). During this lag, a $1\,\mathrm{km}$ TS2 $\mathrm{s}^{-1}$ anomaly at 500 hPa is associated with a $1.91\,\mathrm{km\,s}^{-1}$ anomaly at 100 hPa, with a correlation of 0.46. While the forecast systems capture this behavior qualitatively, most underestimate the magnitude of the correlation and regression for wave 1. For the high-top models, biases identified in the regression coefficients are mirrored by the biases in correlations. However, for the low-top models this is not the case. For example, BoM has one of the highest correlations for wave 1 of any model, while its regression coefficient is the lowest. This is due to the fact that BoM underestimates the variance of wave 1 100 hPa heat fluxes by more than 50 % (Fig. 2a), and this underestimation is likely a reflection of the model's poor simulation of stratospheric variability more generally as documented in Domeisen et al. (2020a). While most high-top models do not show strong biases in the wave 1 100 hPa heat flux variance (Fig. 2a), this may be a case of two biases canceling each other, i.e., too

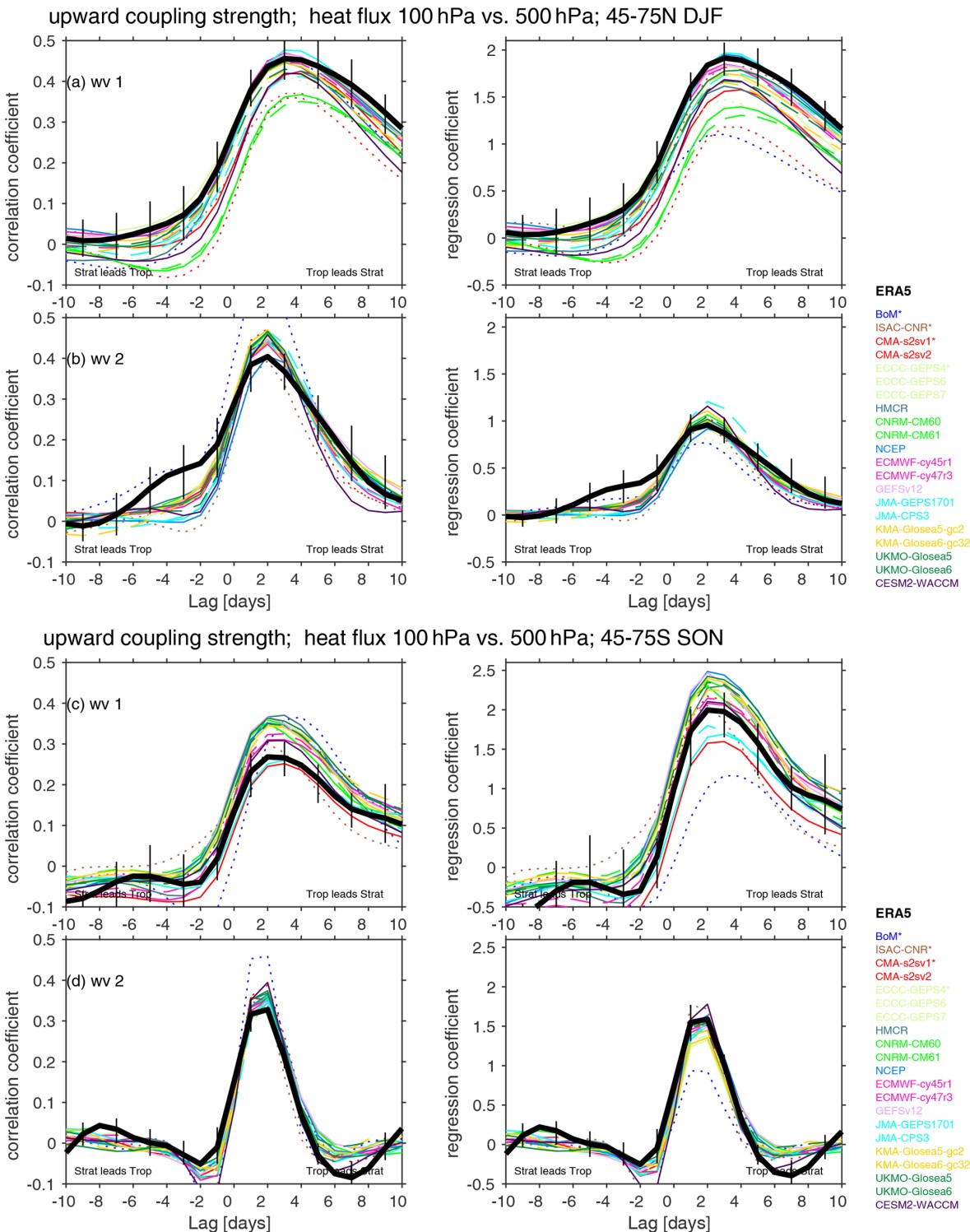

**Figure 3.** Coupling of $\overline{v'T'}$ 45–75° at 500 hPa with that at 100 hPa measured in terms of the correlation coefficient (left column) and regression coefficient (right column). $\overline{v'T'}$ at 500 hPa is taken from days 11–22, and we range $\overline{v'T'}$ at 100 hPa from 10 d prior (i.e., days 1–12) to 10 d after (i.e., days 21–32). Low-top models are dotted. Older versions of high-top models are dashed. Vertical black lines show the range in coupling strength in ERA5 upon subsampling to match each of the 21 S2S forecast systems, and the solid black line indicates the mean of these 21 coupling strengths from ERA5. Panels **(a)** and **(b)** are for the NH and December–January–February, while panels **(c)** and **(d)** are for the SH and September–October–November. Panels **(a)** and **(c)** correspond to wave 1 (wv1), and panels **(b)** and **(d)** correspond to wave 2 (wv2).

much tropospheric wave 1 variability being compensated for by too little upward wave propagation.

The bias in the regression coefficient for each model when compared to the corresponding period in ERA5 is shown in the top row of Fig. 4. Most models suffer from too weak an upward coupling, with only two models (NCEP and JMA CPS3) simulating a stronger regression coefficient than ERA5. The multi-model mean regression coefficient is biased low by 11 % for high-top models and by 25 % for low-top models. Figure 5a further considers the relationship between model biases in coupling strength and vertical lid height by contrasting the bias in coupling with the base-10 logarithm of the vertical lid pressure. While the model lid level is anticorrelated with wave 1 upward-coupling strength ($r = -0.34$, not significant), a more pronounced and statistically significant effect is evident when comparing coupling strength with the magnitude of the climatological wave 1 heat flux in the troposphere (an indication of how well each model represents quasi-stationary waves, $r = 0.5$; Fig. 6a). Models with a better representation of climatological quasi-stationary wave 1 better represent its upward coupling. This effect is even more pronounced if we compare climatological heat flux at 100 hPa with the coupling strength ($r = 0.70$; not shown).

The upgrade of the CMA system from a low-top to high-top model led to a 29 % reduction in its bias in wave 1 upward-coupling strength, while the transition from ECCC GEPS 4 to ECCC GEPS 7 led to a 67 % reduction in its bias (Fig. 4a). Of the high-top models, CNRM struggles the most with the upward-coupling strength, and the upgrade from CNRM-CM6-0 to CNRM-CM6-1 improved the fidelity of the simulation by 21 %. GloSea6 (both KMA and UKMO) improved by 47 % over GloSea5. ECMWF CY47R3 is also improved over its earlier version, though the earlier version was already among the most realistic across all forecast systems, and hence there was less room for improvement.

This overall underestimate of wave 1 upward coupling is confirmed in Fig. 7, which shows the regression coefficient between 500 hPa height anomalies and the wave 1 heat flux at 100 hPa 3 d afterwards for December and January initializations. This analysis is performed for only seven of the models due to data availability and storage constraints. Consistent with previous work (e.g., Garfinkel et al., 2010), low heights in the northwestern Pacific and high heights in the Atlantic sector are associated with pulses of wave 1 heat flux in the lower stratosphere. These anomalies are in phase with climatological wave 1 and thus constructively interfere with it. The models systematically underestimate the regression coefficient in both sectors. The low-top CMA and BoM are particularly biased, again revealing the importance of the model top.

The above results suggest that the S2S forecast systems need a higher model lid and more realistic stationary waves in the troposphere to simulate realistic upward wave 1 coupling between 500 and 100 hPa in the boreal winter. Biases

are smaller for wave 2 upward coupling in the NH winter. Coupling is too strong in 13 of 21 models (Fig. 4a), and the multi-model mean bias is 2.5 % too strong. JMA GEPS1701 simulates a coupling strength 49 % stronger than in ERA5; however in its updated version (JMA CP3) the bias drops to 7.9 %. The mean bias of the regression coefficient is larger for low-top vs. high-top models: specifically, coupling is 9 % too weak in low-top models vs. 1.6 % too strong in the most recent version of high-top models.

BoM suffers from an unrealistically strong correlation (Fig. 3). However, its upward-coupling regression coefficient is the weakest among all models with too weak a bias of 24 %. This apparent paradox is, as before, due to its wave 2 variability at 100 hPa that is too weak. The wave 2 coupling strength is significantly correlated to the model lid ($r = -0.49$, Fig. 5b) and to climatological stationary wave 2 in the lower stratosphere ($r = 0.45$ for climatological $\overline{v'T'}$ at 100 hPa) but not in the troposphere. Finally, the forecast systems better capture the tropospheric precursors of 100 hPa wave 2 heat flux as compared to wave 1 heat flux, with CNRM and UKMO in particular simulating regression coefficients of a reasonable magnitude (Fig. S1 in the Supplement).

In the Southern Hemisphere spring, models systematically have too strong a variance in tropospheric (500 hPa) planetary wave heat flux, with the exception of ISAC-CNR, which underestimates the variance (Fig. 2b). In contrast, lower-stratospheric (100 hPa) planetary wave heat flux is generally too weak in most models. The multi-model mean regression between the 500 and 100 hPa wave 1 heat flux is 6 % too strong (Fig. 3c, d); however there is a large spread among the models (Fig. 4b). High-top models overall perform better: the model lid and regression coefficients are significantly correlated ($r = -0.56$ for both wave 1 and wave 2; Fig. 5g, h). Biases are also smaller in models with better climatological stationary waves (Fig. 6g, h), though this relationship is sensitive to the inclusion of BoM.

## 3.2 The sensitivity of the stratospheric polar vortex to upward wave driving from the lowermost stratosphere

In order for models to fully capture the effect of tropospheric variability on the polar vortex, they must not only capture the upward flux of wave activity from the troposphere to the lower stratosphere but also simulate a reasonable sensitivity of the polar vortex to lower-stratospheric wave activity. We diagnose this sensitivity of the polar vortex by computing the lagged correlation and regression between 10 hPa polar-cap height anomalies and the sum of wave 1 and wave 2 100 hPa heat flux (Fig. 8).

In the Northern Hemisphere, the reanalysis correlation peaks when polar-cap mid-stratospheric heights lag lower-stratospheric heat flux by 7 d, and most models simulate a similar lag (Fig. 8a). Most models underestimate the magni-

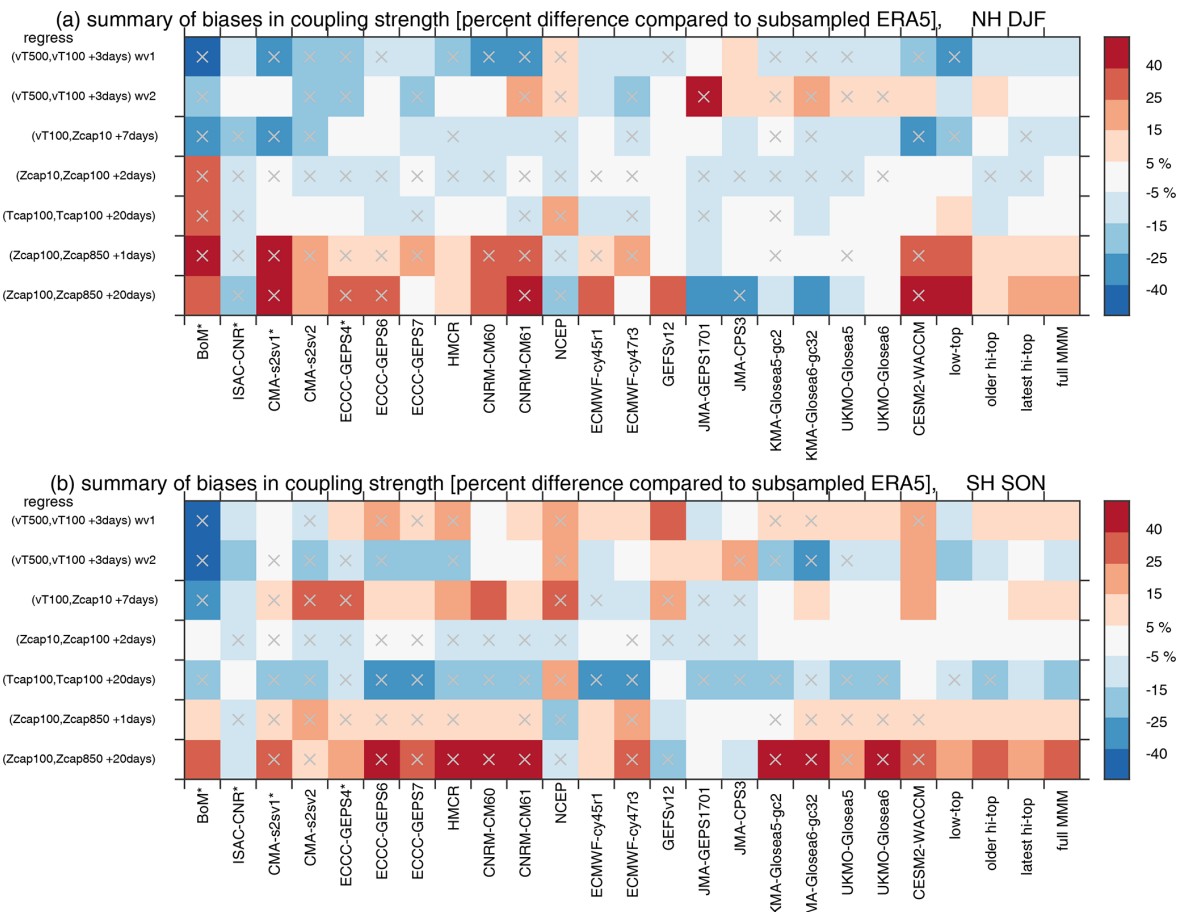

**Figure 4.** Summary of the biases in coupling strength. For each forecast system we compare the coupling strength for the corresponding identical period in ERA5 and then show the percentage error. The bias is defined as the difference in coupling strength between the model and ERA5 divided by the coupling strength in ERA5 for the corresponding dates of each model. A gray × indicates models and metrics for which all ensemble members simulate a bias in the coupling strength of the same sign or, alternatively, if ERA5 does not fall within the envelope of the available members. Low-top models are denoted with an asterisk after their name. Coupling strength is defined using regression, and the analogous figure for correlation is shown in Fig. S2. The top row shows upward coupling between $\overline{v'T'}$ wave 1 at 500 hPa and at 100 hPa with a lag of 3 d (cf. Fig. 3). The second row is like the first row but for wave 2. The third row shows sensitivity of the $Z$ 10 hPa polar cap to 100 hPa heat flux lagged by 7 d (cf. Fig. 8). The fourth row shows coupling strength of the $Z$ 10 hPa polar cap with the $Z$ 100 hPa polar cap with a lag of 2 d (cf. Fig. 12). The fifth row shows persistence of the $T$ 100 hPa polar cap on day 20 (cf. Fig. 14). The sixth and seventh rows show coupling strength of the $Z$ 100 hPa polar cap with the $Z$ 850 hPa polar cap with a lag of 1 and 20 d (cf. Fig. 13).

tude of the coupling, however: the regression coefficient at lag 7 d is too weak in all models except ECCC GEPS 6, with BoM, CMA, and CESM2–WACCM producing particularly large biases (Fig. 4a). This underestimation is pronounced for the low-top models (too weak a bias at 23 % in low-top models vs. 9 % in high-top models). Models with a stronger bias in climatological 500 hPa heat flux suffer from particularly pronounced too weak a sensitivity ($r = 0.82$; Fig. 6c). Similarly, models with a cold-vortex bias also suffer from too weak a sensitivity ($r = 0.53$, Fig. S5c). These effects are more important in explaining intermodel spread than the model lid (Fig. 5c). The models are similarly biased if we contrast 100 hPa heat flux to the tendency of 10 hPa polar-

cap height (e.g., Fig. 7 of Dunn-Sigouin and Shaw, 2015, not shown).

The net effect of the models' (i) underestimation of upward wave propagation from 500 to 100 hPa and (ii) polar vortex that is undersensitive to 100 hPa heat flux is that NH stratospheric polar variability is not coupled strongly enough to tropospheric variability. This is summarized in Fig. 9, which shows maps of the regression coefficient between 500 hPa height anomalies and the tendency in 10 hPa polar-cap heights over a 10 d period, analogous to Fig. 1 of Garfinkel et al. (2010). Low tropospheric heights in the North Pacific and high tropospheric heights over Scandinavia and the Ural mountains precede weakening of the vortex, but the regression coefficients are underestimated by all

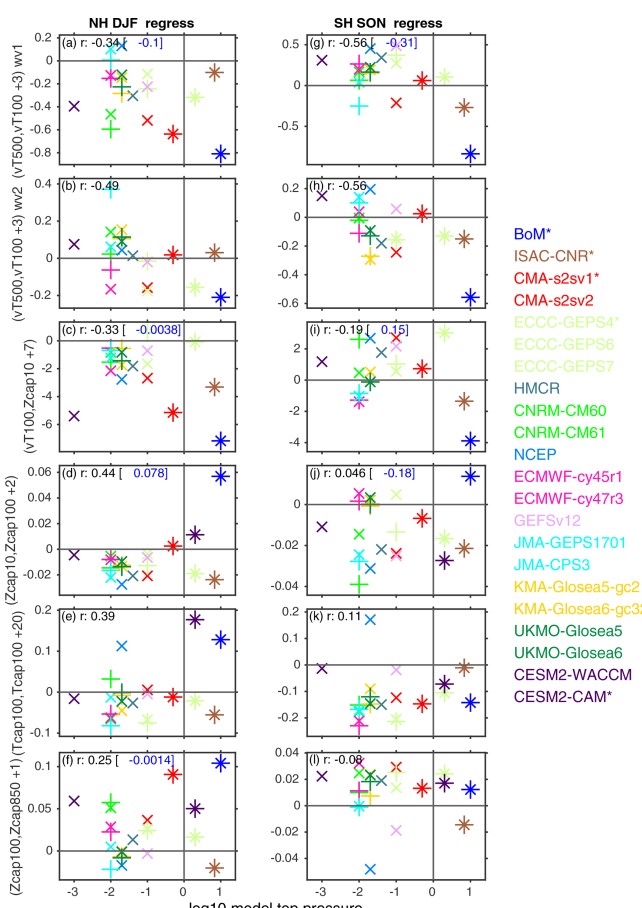

**Figure 5.** Relationship between (*y* axis) the bias in coupling strength as compared to ERA5 for corresponding dates and (*x* axis) the logarithm of the vertical lid for each model. Coupling strength is defined using regression; the corresponding figure for correlations is Fig. S3. The correlation for each panel is indicated, and also the correlation without BoM is in blue if this correlation differs from the overall correlation by more than 0.2. The left column is for the NH in DJF, and the right column is for the SH in SON. Low-top models are indicated by *, older versions of high-top models are indicated by +, and the latest version of high-top models are indicated by ×. **(a, g)** Upward wave 1 coupling on day 3 from Fig. 3, **(b, h)** upward wave 2 coupling on day 3 from Fig. 3, **(c, i)** sensitivity of the $Z10$ hPa polar cap to 100 hPa heat flux on day 7 from Fig. 8, **(d, j)** coupling strength of the $Z10$ hPa polar cap to the $Z100$ hPa polar cap on day 2 from Fig. 12, **(e, k)** persistence of the $T100$ hPa polar cap on day 20 from Fig. 14, and **(f, l)** coupling strength of the $Z100$ hPa polar cap to the $Z850$ hPa polar cap on day 1 from Fig. 13. The lid of GloSea is at 85 km; we represent this with a value of 0.02 hPa. The lid of WACCM is at 140 km; since the levels in the ionosphere are not expected to improve the representation of stratosphere–troposphere coupling, we represent this model with a lid at 0.001 hPa (still the highest lid of any model). A null hypothesis of no relationship can be rejected at the 95 % confidence level using a two-sided Student's *t* test for correlations exceeding 0.42.

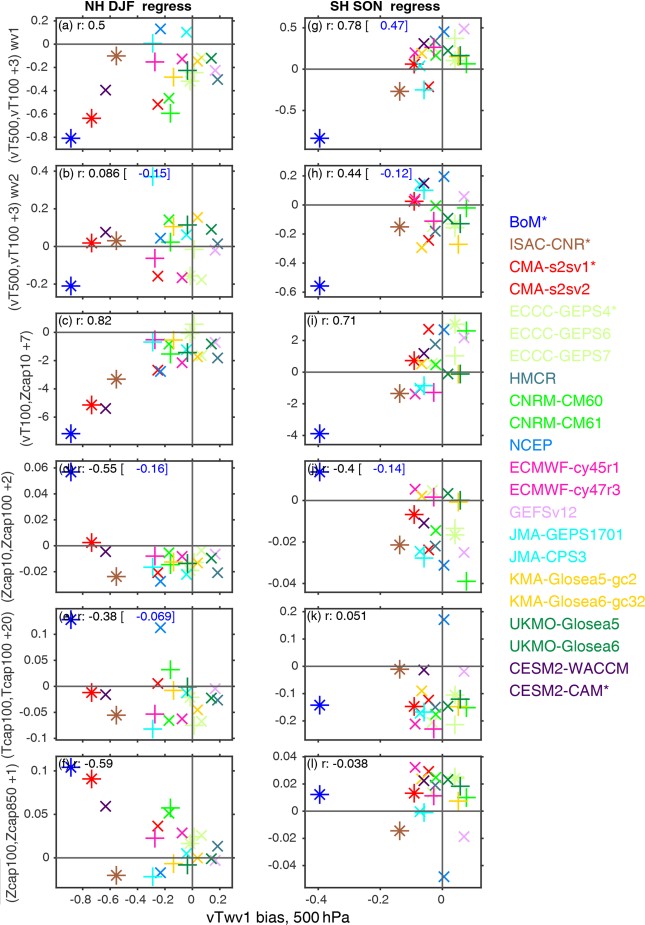

**Figure 6.** As in Fig. 5 but for climatological wave 1 $\overline{v'T'}$ bias for days 22–28 at 500 hPa from 45–75° on the *x* axis (km s$^{-1}$ TS3). The corresponding figure for correlations is in Fig. S4.

models. Note that NCEP is the least-biased model, and this model is the only one which overestimates upward coupling of 500 hPa heat flux with 100 hPa heat flux, although it still has a vortex that is undersensitive to 100 hPa heat flux. The low-top BoM and CMA are the most biased in terms of upward coupling. UKMO and CNRM capture the effect of the Ural high on the vortex, but they underestimate that of the North Pacific low; recall that these models also succeed in simulating the tropospheric precursors of 100 hPa wave 2 heat flux (Fig. S1).

Finally, Fig. 8a shows that there are negative correlation and regression coefficients between polar-cap height and 100 hPa heat flux when polar-cap height leads heat flux. In other words, a stronger polar vortex tends to precede weakened heat flux, while a weaker polar vortex tends to precede strengthened heat flux. This is associated with the polar vortex's ability to regulate its own wave driving (Matsuno, 1970). The models estimate this effect accurately in the multi-model mean (bias less than 3 %). The model which most strongly underestimates this effect is GloSea5 (both

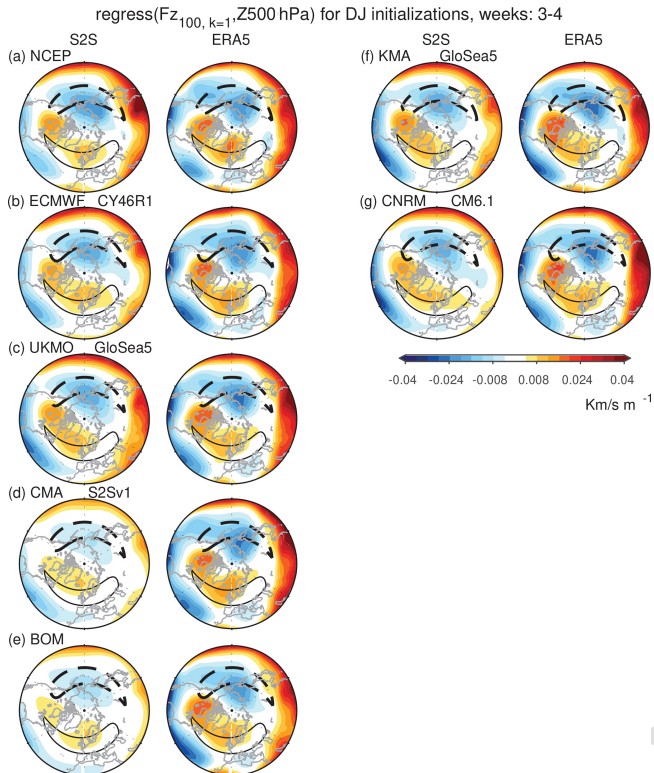

**Figure 7.** Maps of the regression coefficient between $Z500$ anomalies and $v'T'_{100\,\text{hPa},k=1}$ anomalies 2 [TS4] d later using weeks 3 and 4 of December–January initializations. For each model we show the ERA5 subsampled to match each forecast system. Climatological wave 1 of $Z500$ between 45 and 75° N is shown with black contours.

KMA and UKMO); however there is a marked improvement in GloSea6, with biases dropping from 22 % to 8 %. BoM and JMA CP3, on the other hand, overestimate this effect. There is no relationship across models between this effect and either the model lid, the climatological wave 1 strength, or the cold-pole bias.

The sensitivity of the SH polar vortex to 100 hPa extratropical heat flux is overestimated in most models; however the absolute error is higher for low-top models (Figs. 4b and 8b). The ability of the vortex to modulate its own wave driving is less pronounced in the SH than in the NH; however the models underestimate this effect by 28 % (Fig. 8).

### 3.3 Biases in interannual variance of daily heat flux extremes

Section 3.1 and 3.2 demonstrated that there are systematic biases in heat flux variance and the associated upward coupling at subseasonal timescales. This bias also extends to a poor simulation of interannual variability in daily heat flux extremes. We quantify this behavior by computing the 95th percentile of daily eddy heat fluxes (wavenumbers 1–3) for each winter of the 1999–2010 period (Figs. 10–11). The median

(marker) and range of 2 standard deviations (whiskers) of those values for each lead time are shown at 50 and 300 hPa. This analysis thus shows year-to-year spread in the highest heat flux extremes. An equivalent analysis was done for the 5th-percentile (lowest) extremes with qualitatively similar results (not shown).

For the NH (Fig. 10), the interannual spread in positive heat flux extremes at 50 hPa becomes dramatically reduced for most systems after week 1 compared to the reanalysis. In other words, the year-to-year variations in stratospheric heat flux extremes are not well captured in the S2S forecast systems beyond a week. This contrasts with the behavior at 300 hPa, where most forecast systems capture the reanalysis interannual spread in extremes through week 4 (days 22–28), though there is some reduction in spread by week 5. One exception to the generally well-captured positive heat flux extremes at 300 hPa is BoM: in BoM these extremes are persistently too low at both 300 and 50 hPa.

For the SH (Fig. 11), the systems underestimate the interannual spread of daily heat flux extremes beyond week 1 at 50 hPa and beyond week 2 at 300 hPa. This reduction in the spread of the positive heat flux extremes is particularly evident at 50 hPa, despite most systems capturing the median values of the 95 % percentile extremes well (except for BoM which underestimates the median extreme value after week 1 and WACCM which underestimates the median after week 4). At 300 hPa, most systems show a reduction in the interannual spread of positive heat flux extremes at weeks 3–5 compared to the reanalysis spread (and systematically underestimate the median extreme value). This attenuation in the spread of extreme values is thus more evident in the SH troposphere compared to the NH troposphere.

These analyses suggest that at long lead times, the models' daily heat flux extremes either are less sensitive to or lack external sources of interannual variability that arise due to, e.g., teleconnections, or are missing certain internal processes that lead to variability on longer timescales. While this reduction in spread is also apparent for median values of heat fluxes in some models (not shown), it is much weaker, suggesting that the extremes of the eddy heat flux distribution are more sensitive to this bias than the median.

### 3.4 Downward propagation of stratospheric polar vortex anomalies within the stratosphere

The downward propagation of mid-stratospheric polar vortex anomalies to the lower stratosphere is considered in Fig. 12, which shows the lagged correlation of 10 hPa polar-cap height with 100 hPa polar-cap height. In the NH, downward propagation peaks after 2 d in ERA5 and in most models. While several models simulate this downward propagation realistically, there is a systematic underestimation of the magnitude of downward coupling within the stratosphere (Fig. 4a, fourth row), with only 2 of 21 systems (low-top BoM and CMA S2Sv1 [CE5]) simulating too strong a coupling

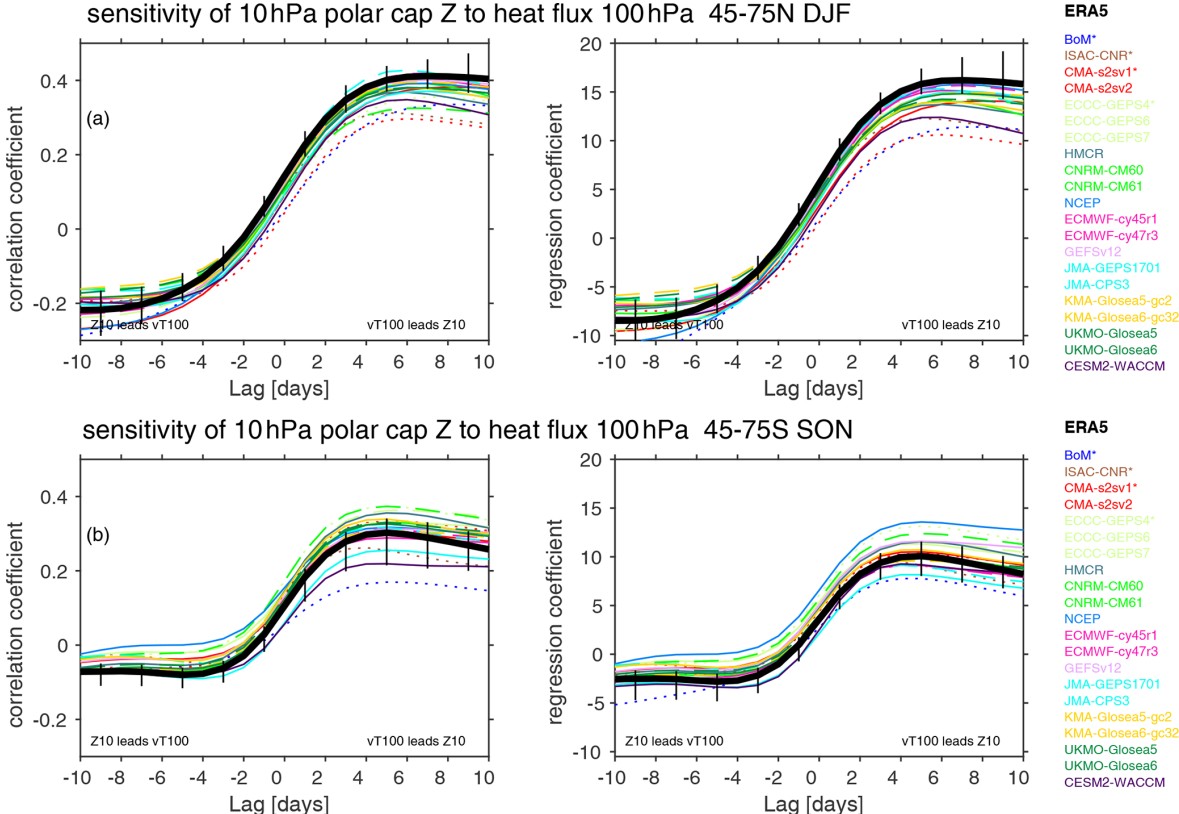

**Figure 8.** Sensitivity of the 10 hPa polar cap $Z$ to $\overline{v'T'}_{k=1+2}$ at 100 hPa for 45–75°. $\overline{v'T'}_{k=1+2}$ at 100 hPa is taken from days 11–22, and we range 10 hPa polar cap $Z$ from 10 d prior to 10 d after. Low-top models are dotted. Older versions of high-top models are dashed. The SH $\overline{v'T'}$ is multiplied by −1 before the analysis is performed to allow for a simpler comparison to the panels for the NH. Vertical black lines show the range in coupling strength upon subsampling the ERA5 reanalysis to match each of the forecast systems, and the solid black line indicates the mean of these coupling strengths.

strength. Biases are even more pronounced for lags of 5 to 10 d, though it is smaller for 20 d lags aside from low-top models (Fig. 12). BoM again shows the largest bias (specifically, an overestimation of coupling strength), even if its correlation indicates an underestimation of coupling strength; this is again a reflection of a poor simulation of stratospheric variance (Fig. 2a). There is a notable improvement from CNRM-CM6-0 to CNRM-CM6-1, from UKMO GloSea5 to UKMO GloSea6, and in successive versions of ECCC GEPS, though not from ECMWF CY45R1 to ECMWF CY47R3 or JMA GEPS1701 to JMA CP3 (Fig. 4a). Low-top models (Fig. 5d) and models with relatively poor climatological stationary waves tend to simulate a stronger downward-coupling strength; however, this relationship is dominated by a single outlier model (BoM). If the correlation is computed without this model, there is instead no detectable relationship between downward-coupling strength and either model lid height or stationary wave amplitude.

Similar to the NH, downward coupling of polar-cap height from the middle to lower stratosphere is too weak in the SH in nearly all models (Figs. 12b, 4b), especially at longer lags. Notably, in the SH the reanalysis relationship actu-

ally strengthens between days 4–20, which may be related to chemistry–circulation coupling in austral spring, as discussed by Simpson et al. (2011). High-top models overall perform better and have a lower absolute error.

### 3.5 Persistence of the polar vortex signal in the lower stratosphere and downward propagation from the lower stratosphere to the near surface

After the stratospheric signal reaches the lower stratosphere, it can subsequently impact the tropospheric circulation. We evaluate whether the models successfully capture this effect using both a regression/correlation approach and a compositing approach.

#### 3.5.1 Regression/correlation perspective on downward-coupling biases

We begin with a regression/correlation approach in Fig. 13a, which shows the lagged regression of 100 hPa polar-cap height with 850 hPa polar-cap height in the NH. For lags of less than a week, too strong a bias exceeding 5 % is

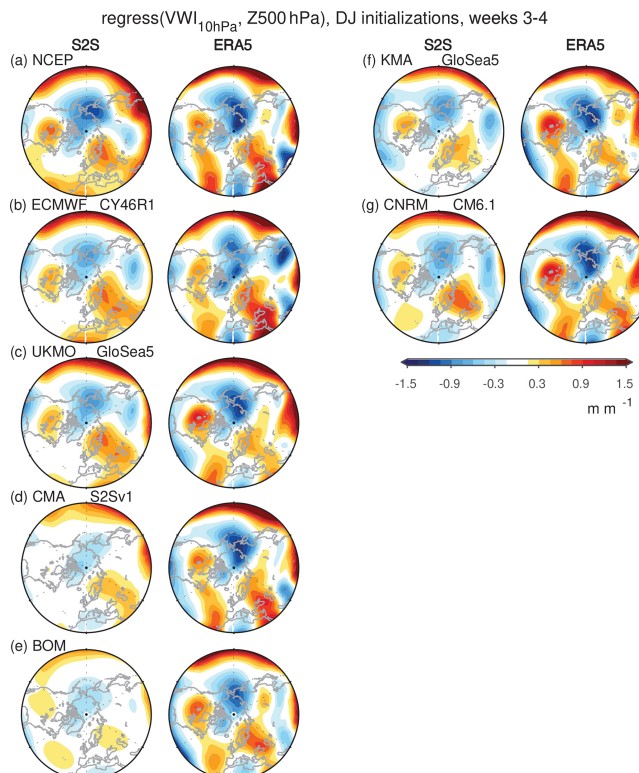

regress(VWI$_{10hPa}$, Z500 hPa), DJ initializations, weeks 3–4

**Figure 9.** Regression coefficient of the change in polar-cap geopotential height at 10 hPa over 10 d (Zcap at day 10 minus Zcap at day 0; a vortex weakening index) with Z500 anomalies on day 0, using weeks 3 and 4 of December–January initializations for Z500 anomalies on day 0.

evident in 12 models, while too weak a bias exceeding 5 % is evident in only three models (ISAC-CNR, JMA GEPS1701, and NCEP). Too strong a downward coupling for nearly simultaneous lags is consistent with Kolstad et al. (2020) for ECMWF. For later lags, additional models develop biases that are too weak, and individual models suffer from large biases. For example, CESM2–WACCM, CESM2–CAM, and CNRM (both generations) overestimate the coupling, while JMA (both generations) and NCEP underestimate it. There is a substantial improvement from ECMWF CY45R1 to ECMWF CY47R3 and from ECCC GEPS 6 to ECCC GEPS 7, but we see no evidence of an improvement from the other modeling centers.

Compared to other stratosphere–troposphere coupling metrics (Fig. 4a), this part of the coupling process is the most consistently biased (in an absolute sense) across models. The bias is less evident upon examining the correlation (Figs. 13, S2), likely because some of these models also suffer from too strong a bias in the variance for 850 hPa geopotential height (Fig. 2a): too strong a regression coefficient combined with too strong a variance can lead to a reasonable net correlation. These smaller biases for a correlation approach are consistent with Lee and Charlton-Perez (2024) for models which over-

lap with those considered here (ECMWF CY45R1, CNRM-CM6-0, UKMO GloSea5, and NCEP). Downward coupling is too strong in models with overly weak climatological tropospheric wave 1 (Fig. 6f); this relationship is consistent with the documented effect of planetary waves dampening synoptic-eddy feedback (i.e., Feldstein and Lee, 1998; Lorenz and Hartmann, 2003), though the full range of interactions of planetary waves with vortex perturbations is still not fully understood (Song and Robinson, 2004; Domeisen et al., 2013; Hitchcock and Simpson, 2014; Garfinkel et al., 2023).

The downward-coupling signal in later weeks is potentially related to the persistence of lower-stratospheric vortex anomalies, as the slow radiative decay of these anomalies allows for lower-stratospheric variability to affect surface climate on subseasonal timescales (Hitchcock and Simpson, 2014). Specifically, if polar vortex anomalies were to decay too fast, then this could lead to too weak a downward coupling at later lags. This possibility is examined in Fig. 14, which shows the lagged autocorrelation of polar-cap temperature at 100 hPa; we focus here on temperature rather than geopotential height due to its close connection with radiative timescales and tracer concentrations. Three models simulate biases of the autocorrelation of polar-cap temperature at 100 hPa on day 20 exceeding 5 % (low-top CESM2–CAM and BoM and high-top NCEP). Seventeen other models simulate overly fast decay if we subsample ERA5 to match the dates actually used for each model (Fig. 4). The overly fast decay exceeds 10 % only for ECCC GEPS 6, ECCC GEPS 7, CNRM-CM6-1, and JMA GEPS1701 and is more pronounced (though not statistically significantly so) in models with higher tops and better stationary waves (Figs. 5e and 6e). The correlation between (i) the autocorrelation of the polar-cap temperature at 100 hPa on day 20, with (ii) the regression of the 100 hPa polar-cap height with the 850 hPa polar-cap height on day 20, is 0.34 such that a stronger autocorrelation of polar-cap temperature is associated with a strong surface signal. This relationship is somewhat weaker than the corresponding relationship with tropospheric stationary waves ($r = -0.45$).

In the SH, downward coupling of the polar-cap height from the lower stratosphere to the surface is too strong in most models for both nearly simultaneous lags and at 20 d lags (Figs. 13b, 4b). Two models (NCEP and GEFSv12) suffer from too weak a coupling bias for simultaneous lags exceeding 10 %, and for later lags ISAC-CNR and JMA CPS3 also simulate coupling that is too weak. For nearly all other models, however, overly strong downward coupling occurs even as polar-cap temperature anomalies decay too fast in these models (Figs. 4b, 14b). Hence too strong a downward coupling likely reflects overly strong eddy feedback, as has been recently shown explicitly for a subset of these models (Garfinkel et al., 2024). Consistent with this, too strong a coupling bias is more pronounced at later lags than nearly simultaneous lags (Fig. 4b).

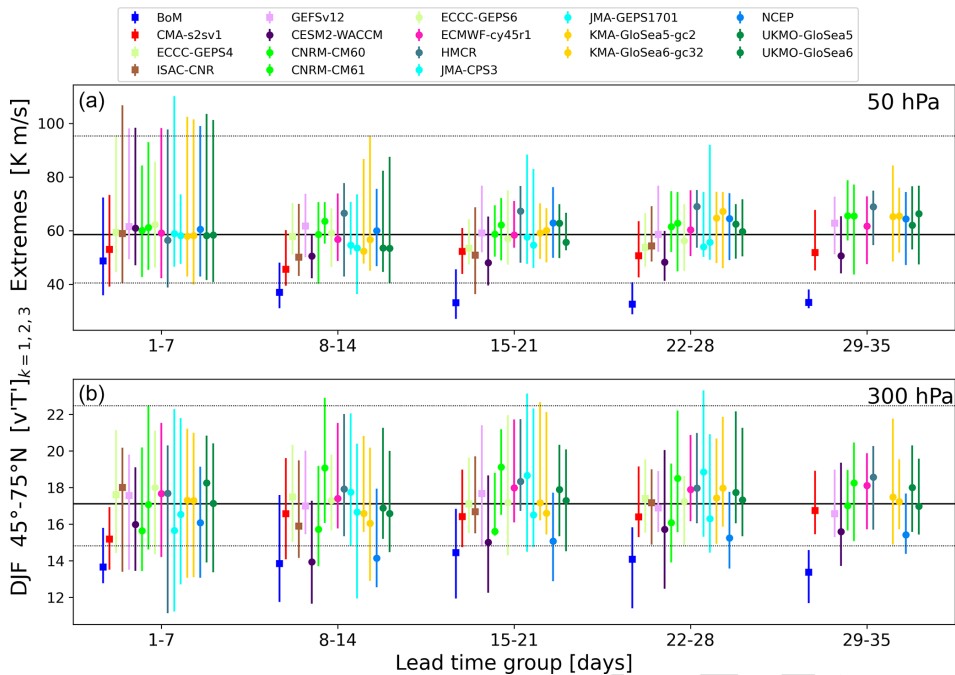

**Figure 10.** The 95th-percentile daily eddy heat flux extremes for 45–75° N during winter (DJF) from 1999–2010 for all models by weekly lead time group. The median is indicated by the marker and the ±2 standard deviations by the whiskers for **(a)** 50 hPa and **(b)** 300 hPa. The equivalent values from the reanalysis are given by the horizontal black lines (bold: median; thin: ±2 standard deviations).

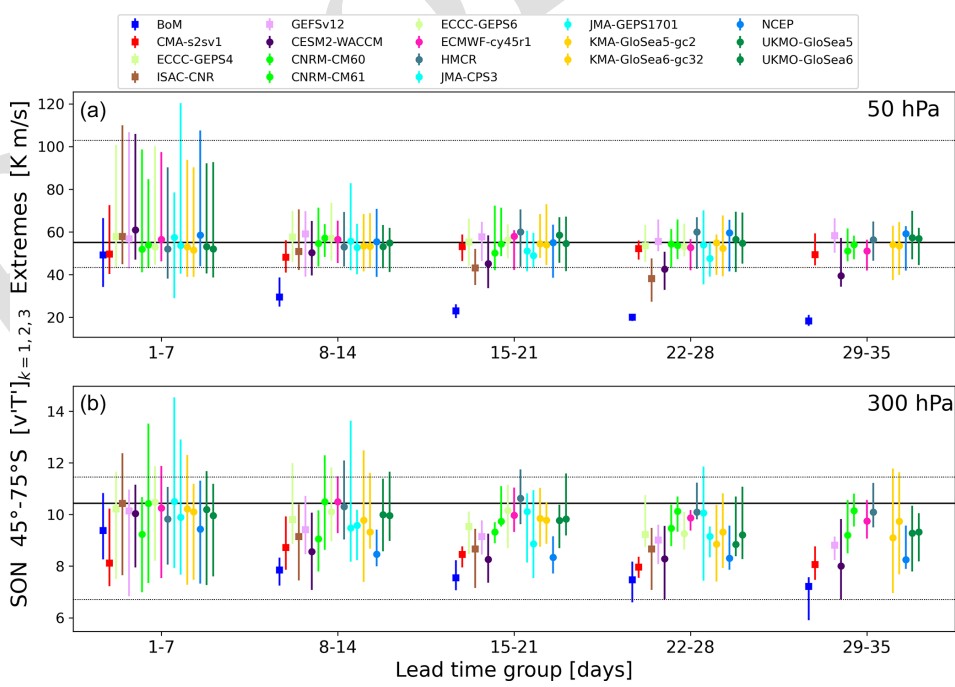

**Figure 11.** The 95th-percentile daily eddy heat flux extremes for 45–75° S during spring (SON) from 1999–2010 for all models by weekly lead time group. The median is indicated by the marker and the ±2 standard deviations by the whiskers for **(a)** 50 hPa and **(b)** 300 hPa. The equivalent values from the reanalysis are given by the horizontal black lines (bold: median; thin: ±2 standard deviations).

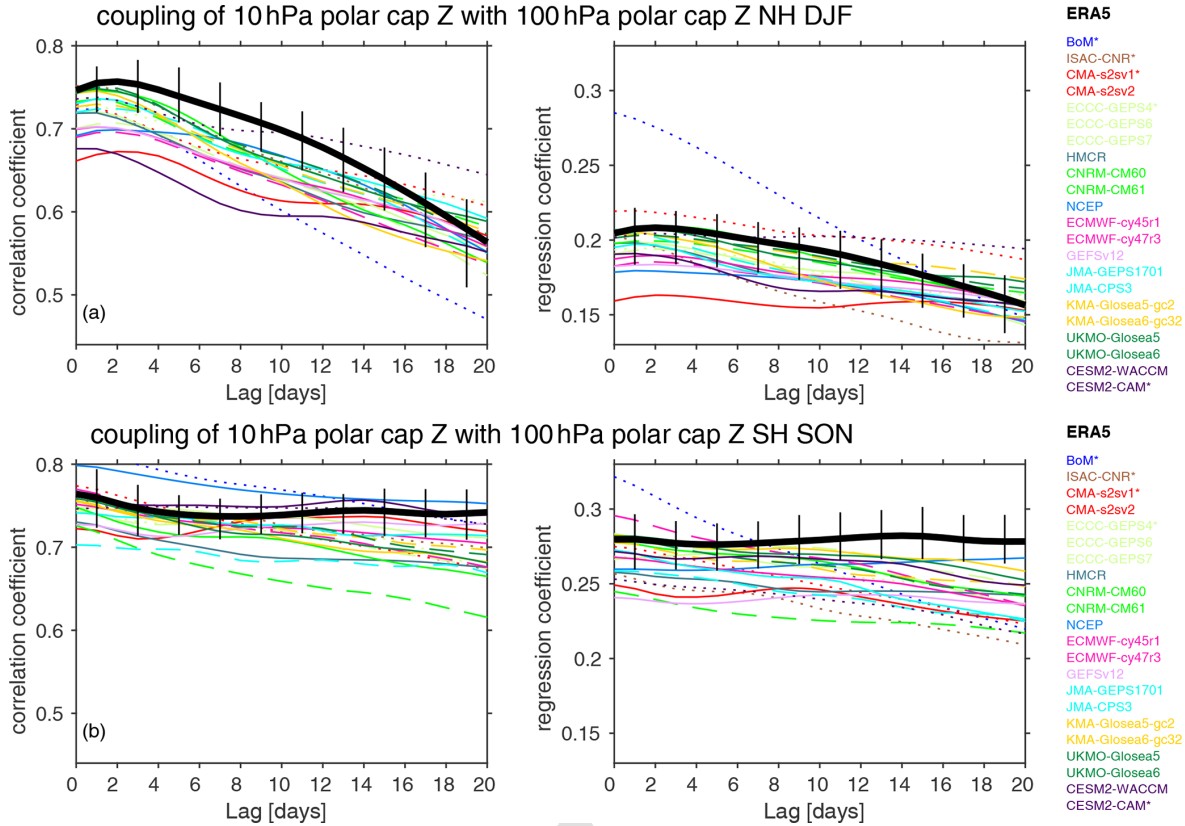

**Figure 12.** Coupling of the polar-cap height at 10 hPa with that at 100 hPa. Polar cap $Z$ at 10 hPa is taken from days 9–12, and we show the range of 100 hPa polar cap $Z$ with simultaneous data to Z10 20 d after. Low-top models are dotted. Older versions of high-top models are dashed. Vertical black lines show the range in coupling strength upon the subsampling ERA5 reanalysis to match each of the forecast systems, and the solid black line indicates the mean of these coupling strengths.

### 3.5.2 Extreme stratospheric events' perspective on downward-coupling biases

So far our consideration of downward coupling has been based on a correlation/regression analysis. This analysis does not explicitly consider the role of extreme events of the stratospheric polar vortex for surface predictability. Specifically, a highly disturbed or extremely strong polar vortex has stronger impacts than more typical vortex variability; for example, White et al. (2020, 2022) and Garfinkel et al. (2023) find that the near-surface response scales linearly with the lower-stratospheric perturbation. We now consider whether the S2S systems capture downward coupling for these extreme events.

We quantify the biases in downward impact by forming composites of initializations in which polar-cap height anomalies at 10 hPa exceed 500 m (strong vortex) or are more negative than −500 m (weak vortex) on day 10 and compute the Zcap at 100 hPa on days 10–31 (Fig. S5). These thresholds are chosen to consider extreme conditions only (approximately 9 % of all available members are included in each composite), though results are similar for a threshold of, say,

±400 m (not shown). The biases averaged from days 20–30 are summarized in Fig. 15a. For both the SH and NH, many more models simulate too weak a downward propagation within the stratosphere than one that is too strong. This effect is consistent with the regression coefficients (Figs. 4 and 12). The bias is particularly pervasive for weak vortex events.

Next, we consider biases in the downward coupling of extreme vortex events between 100 hPa and the near CE7 surface. Specifically, we form composites of initializations in which polar-cap height anomalies at 100 hPa exceed 175 m (strong vortex) or are more negative than −175 m (weak vortex) on day 10 and plot the Zcap at 850 hPa on days 10–31 (Fig. S6). These thresholds are chosen such that ∼ 7.6 % of all available members are included in each composite, though results are similar for a threshold of, say, ±100 m (not shown). The biases averaged from days 20–30 are summarized in Fig. 15b. In contrast to the downward propagation that is too weak within the stratosphere, most models simulate downward coupling from the lower stratosphere to the near surface that is too strong. There are notable exceptions, however, in both hemispheres. In the NH, ISAC-CNR, both

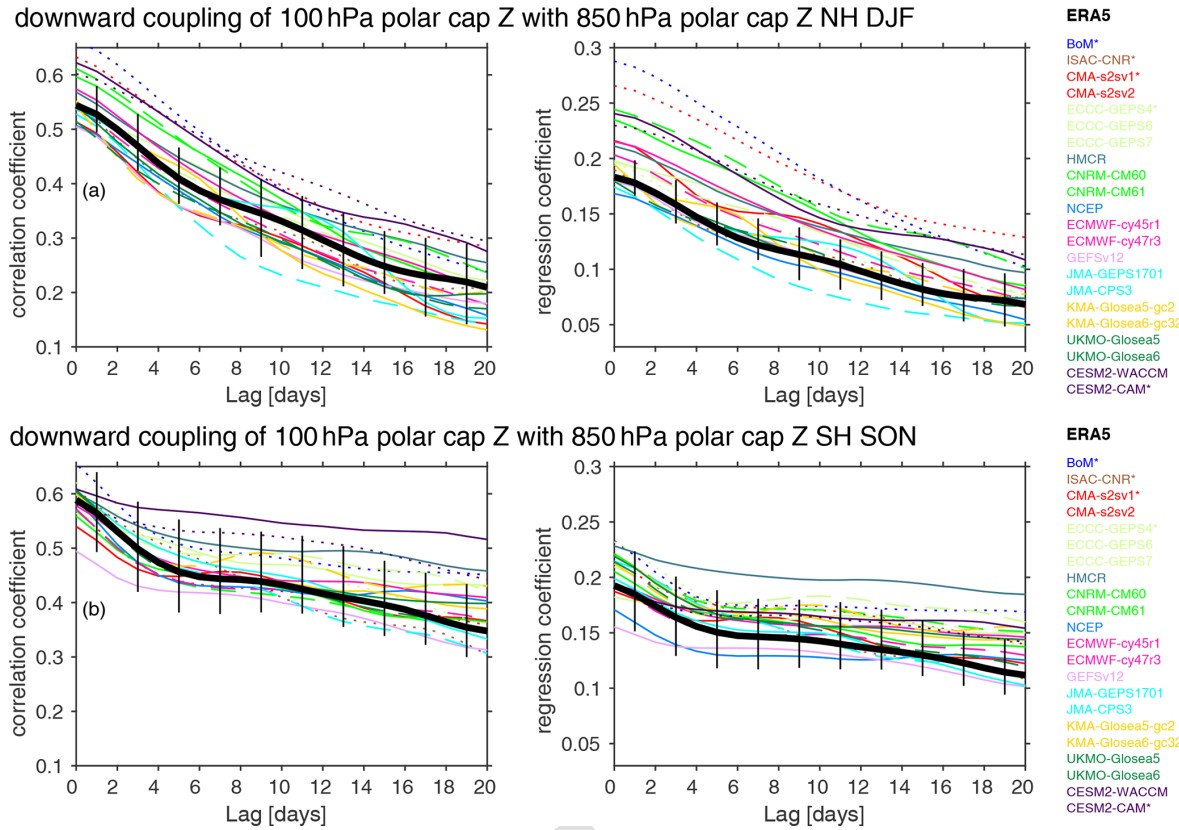

**Figure 13.** Lagged correlation/regression coefficient of the 100 hPa polar-cap geopotential height with that at 850 hPa. Polar-cap geopotential height at 100 hPa is selected from days 9–12 after initialization, and we show the range of 850 hPa polar cap $Z$ simultaneous with $Z$100 up to 20 d later. Low-top models are dotted. Older versions of high-top models are dashed. Vertical black lines show the range in coupling strength upon subsampling the ERA5 reanalysis to match each of the forecast systems, and the solid black line indicates the mean of these coupling strengths.

JMA configurations, and all four GloSea configurations are relatively less biased, consistent with the regression coefficients that are too weak evident in Fig. 4. Similarly, in the SH, ISAC-CNR, NCEP, GESFv12, and both JMA configurations simulate too weak a coupling, which is also consistent with Fig. 4. Overall, downward coupling is too weak within the stratosphere but too strong from the lower stratosphere to the near surface. These two biases tend to compensate each other when considering downward coupling from the mid-stratosphere to the near surface. Indeed, a similar figure but for the Zcap 850 hPa response to extreme events of Zcap at 10 hPa shows lower biases (less than 3 % and 9 % in the SH and NH, respectively) than for Zcap at 100 hPa.

## 3.6 Other possible contributors to intermodel spread in coupling strength

Throughout the text we discussed the role of model lid height and of biases in the stationary waves for biases in coupling processes. These S2S models are also known to suffer from a cold-pole bias in the lowermost stratosphere (Lawrence et al., 2022), and we have explored whether intermodel spread in

the magnitude of this cold-pole bias might be related to spread in the strength of coupling. Figures S7 and S8 consider this possibility; however, we find that its role is weaker than those of lid height or stationary waves for all coupling processes.

An additional possibility is that the number of levels in the troposphere, near the tropopause, or in the stratosphere might be related to the model spread in coupling processes. We evaluate these possibilities with Figs. S9, S10, and S11, respectively. Namely, we contrast coupling strength with the number of model levels below 300 hPa, between 300 and 100 hPa, and between 100 and 10 hPa. The number of tropospheric levels is robustly associated with improved coupling of wave 1 heat flux from 500 to 100 hPa (Fig. S9a), and the correlation (0.67) is stronger than for any of the other factors explored in this paper (e.g., model lid, stationary wave climatology, cold-pole biases). A similar albeit weaker relationship is evident if we consider the number of levels from 300 to 100 hPa (Fig. S10a, $r = 0.44$). This sensitivity is not evident in the SH or for wave 2, however. More levels in the troposphere and near the tropopause are associated with a

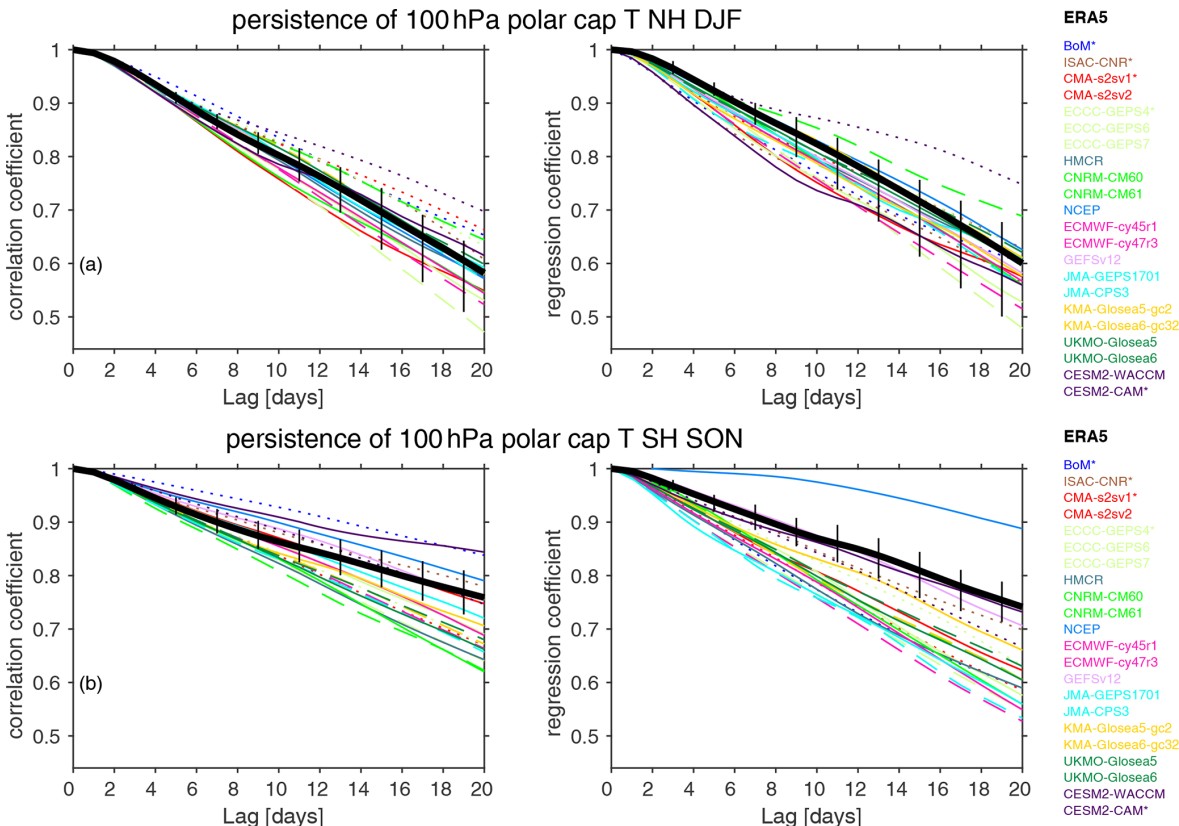

**Figure 14.** Persistence of 100 hPa polar cap $T$. Polar cap $T$ at 100 hPa is taken from days 9–12, and we then compute its lagged correlation up to 20 d later. Low-top models are dotted. Older versions of high-top models are dashed. Vertical black lines show the range in coupling strength upon subsampling the ERA5 reanalysis to match each of the forecast systems, and the solid black line indicates the mean of these coupling strengths.

more reasonable sensitivity of the polar vortex to incoming wave flux in the NH (Figs. S9c, S10c); however this effect is weaker than the corresponding effect if we consider the climatological heat flux bias (Fig. 6c). There is little relationship between downward propagation within the stratosphere and the number of levels if BoM is excluded (Figs. S9d and j, S10d and j, and S11d and j); however downward propagation from the lowermost stratosphere to the near surface in the NH is better simulated in models with more tropospheric levels (Fig. S9f). Finally, the number of levels between 100 and 300 hPa (i.e., better resolution in the tropopause region) is robustly related to better relaxation timescales for Tcap100 in the Northern Hemisphere (Fig. S10e). Such a relationship makes sense if these relaxation timescales are dictated by poor representation of the transport of water vapor (Riese et al., 2012; Charlesworth et al., 2023).

For essentially all of the coupling processes the number of model levels between 100 and 10 hPa is not significantly correlated to biases in the coupling processes (Fig. S11). This suggests that better simulating/resolving tropospheric and lower-stratospheric processes is the key to reducing some of the biases in the coupling that we are seeing, rather than

getting extra-high resolution higher in the stratosphere. We have tested this possibility by re-calculating the metrics in Fig. 5 but for high-top models only and found that there is still a tendency for higher high-top models to better represent upward wave 2 (not shown). However, for other metrics the correlations are reduced or even change sign, suggesting that once the lid is sufficiently high, the effect of lid height becomes saturated.

## 4 Discussion and conclusions

Variability in the extratropical stratosphere and troposphere are coupled (Baldwin and Thompson, 2009; Kidston et al., 2015). A large pulse of planetary waves in the troposphere can disturb the polar stratospheric vortex, while vortex extremes influence surface climate and extremes for weeks to months afterwards (Domeisen and Butler, 2020). This coupling can potentially provide windows of opportunity for prediction on subseasonal-to-seasonal (S2S) timescales (Butler et al., 2019; Domeisen et al., 2020b); however model biases in either the troposphere or stratosphere can degrade the representation of these coupling processes.

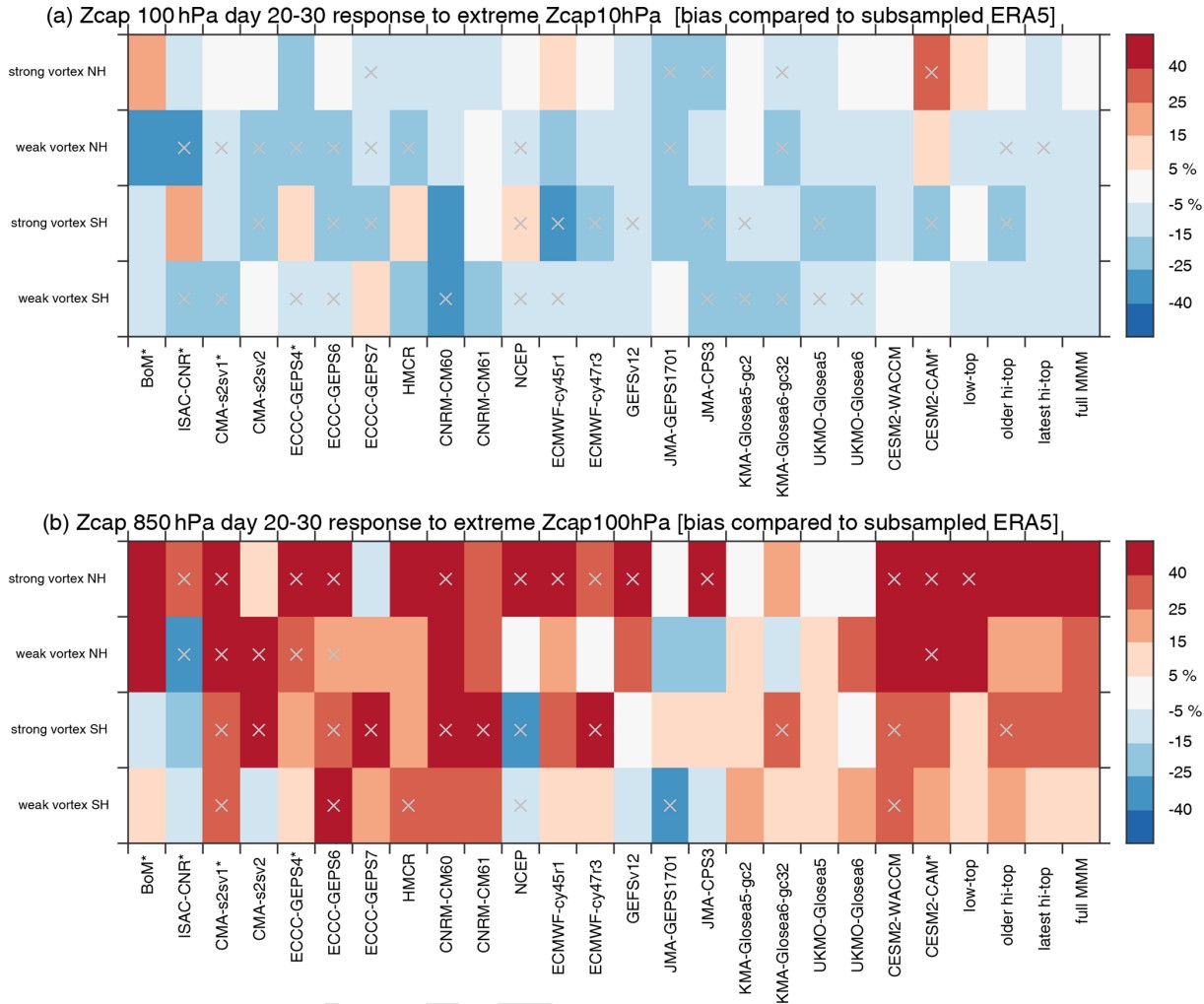

**Figure 15.** Summary of the biases in downward-coupling strength for extreme stratospheric events. We compare each forecast system to the corresponding identical period in ERA5 and then show the percentage error. The bias is defined as the difference between the model and ERA5 divided by the response in ERA5. A gray × indicates models and metrics for which all ensemble members simulate a bias in the coupling strength of the same sign or, alternatively, if ERA5 does not fall within the envelope of the available members. Low-top models are denoted with an asterisk after their name. **(a)** Downward coupling within the stratosphere. The first and third rows show the composite mean of Zcap at 100 hPa for days 20–30 for initializations in which Zcap at 10 hPa on day 10 exceeds 500 m. The second and fourth rows are like the first and third rows, but Zcap at 10 hPa on day 10 is more negative than −500 m. **(b)** Downward coupling from the lower stratosphere to the troposphere. The first and third rows show the composite mean of Zcap at 850 hPa in days 20–30 for initializations in which Zcap at 100 hPa exceeds 175 m on day 10. The second and fourth rows are like the first and third rows, but Zcap at 100 hPa on day 10 is more negative than −175 m. The thresholds lead to ∼ 8.7 % of all available members being chosen averaged across all models, hemispheres, and composites; the mean composite size is 248.

We have performed a comprehensive intercomparison of biases in extratropical stratosphere–troposphere coupling processes in subseasonal forecast systems, with a core focus on systems that contribute to the S2S database (Vitart et al., 2017). We broke up this coupling into six processes that can be diagnosed with a few key metrics in the hopes that they can be easily adopted by modelers to assist ongoing development. Our main results can be summarized as follows:

1. *Upward flux of wave activity to the lower stratosphere*. For the NH, the forecast systems systematically under-

estimate the upward coupling of wave 1 from the mid-troposphere to the lower stratosphere. In contrast, upward coupling of wave 2 is better simulated (Fig. 3, top two rows of Fig. 4a). Upward coupling is better captured in high-top models and, even more robustly, in models with a better representation of climatological quasi-stationary waves (Figs. 5a, 6a) and better tropospheric resolution (Fig. S9). Models underestimate the sensitivity of lower-stratospheric wave 1 heat flux to tropospheric variability in the northwestern Pacific and Euro-

Atlantic (Fig. 7). This relatively better performance for wave 2 as compared to wave 1 is remarkable given the overall poorer performance of these models with respect to the prediction of SSW events dominantly driven by wave 2 (Taguchi, 2018; Domeisen et al., 2020b). This difference between wave 1 and wave 2 biases in the upward wave flux is likely a reflection of the fact that climatological wave 2 heat flux is better represented (and indeed, too strong) in many of these models, while climatological wave 1 is too weak (Fig. S12). However, it is possible that there are additional biases in wave 2 ahead of extreme vortex events.

For the SH, the high-top forecast systems systematically overestimate the upward coupling of wave 1 from the mid-troposphere to the lower stratosphere (Fig. 3, top row of Fig. 4b), in contrast to the underestimation in the NH. Note that the models also better capture climatological wave 1 in the SH than in the NH (Fig. S12), and the intermodel spread in upward coupling in the SH is also linked to each model's representation of climatological wave 1 (Fig. 6g). The tropospheric heat flux variances are systematically too high, while the stratospheric variances are too low, so the relatively successful coupling strength may be due to some kind of cancellation effect (variability in the troposphere that is too high is overcompensating for what would be upward coupling that is too weak).

2. *Sensitivity of the vortex to upward flux of wave activity in the lower stratosphere.*
   For the NH, the polar vortex is not sensitive enough to upward-propagating wave flux (Fig. 8a). This effect is especially pronounced in models with large biases in climatological 500 hPa heat flux (Fig. 6c).
   For the SH, multi-model mean biases are small (Fig. 8b). The intermodel spread is mostly accounted for by the climatological 500 hPa heat flux (Fig. 6i). Note that the forecast systems simulate climatological 500 hPa heat flux better in the SH than in the NH in the multi-model mean (Fig. S12).

3. *Interannual variance of daily heat flux extremes.*
   In both the NH and SH stratosphere, the interannual spread in positive eddy heat flux extremes is strongly reduced for most systems after week 1. This is also evident in the SH troposphere for weeks 3–5 (Figs. 10 and 11). More work is needed to understand (a) what drives this lack of interannual variability in heat flux extremes (one possibility might be poor simulation of teleconnections arising from, e.g., the El Niño–Southern Oscillation, ENSO; Garfinkel et al., 2022; Bayr et al., 2019; Williams et al., 2023); (b) the asymmetry in behavior between the NH and SH troposphere; and (c) the extent to which this bias affects stratospheric circulation extremes, their predictability, and subsequent downward

coupling. Potential implications for subseasonal forecasting include, for example, a failure of the S2S systems to forecast stratospheric heat flux extremes beyond week 1 that are associated with potentially predictable sources of interannual variability.

4. *Downward propagation within the stratosphere.*
   For the NH, there is a systematic underestimation of the magnitude of downward coupling within the stratosphere both when using a regression/correlation approach (Figs. 4a, 12a) or a compositing approach focused on the extreme events (Fig. 15). We were unable to identify any factor that is robustly linked to the intermodel spread in this underestimation (Figs. 5d, 6d, S7d–S11d).
   For the SH, similar to the NH, downward coupling of polar-cap height from the middle to lower stratosphere is too weak in the SH in nearly all models (Figs. 4b, 12b), especially at longer lags; however the biases are generally small ($< 10\,\%$). This finding is confirmed using a composite approach based on extreme events (Fig. 15). As for the NH, we were unable to identify any factor that is robustly linked to the intermodel spread in this underestimation (Figs. 5j, 6j, S7j–S11j).

5. *Persistence of the polar vortex signal in the lower stratosphere.*
   For the NH, the multi-model mean bias for high-top models is less than $5\,\%$; however there is a wide spread across models with persistence that is too strong for some models and decay that is too fast (albeit relatively weak) for most models. We have examined whether intermodel spread in this bias is related to mean-state biases in polar-cap temperatures; however the relationship was weak (Figs. S7, S8). The intermodel spread in this underestimation is not related to the model lid or stationary wave climatology either (Figs. 5e, 6e); however it is related to the number of vertical levels between 100 and 300 hPa (Fig. S10e).
   For the SH, temperature anomalies decay too fast. Models with more levels between 100 and 300 hPa tend to suffer from this problem more severely (Fig. S10k), suggesting that adding resolution is not a panacea. The intermodel spread in this bias is related to mean-state biases in polar-cap temperatures: models with larger mean-state cold biases simulate a better autoregression (Figs. S7, S8). Possible speculative causes for this include (i) a stronger time-mean vortex that can better duct away incoming waves and hence is less variable; (ii) a cold bias that will lead to less efficient longwave emission and hence a slower relaxation back to climatology in response to a temperature anomaly (regardless of sign); and (iii) a third, as yet unknown, bias that may also be important. An additional possibility is that ozone coupling may be crucial for temperature persistence in the SH; however ozone is prescribed to climatological

values in many subseasonal forecasting models. It is notable that NCEP is the only model overpredicting persistence in the SH and that it is one of the few models used in this study that uses prognostic ozone. Additional work is needed to better understand this possibility.

6. *Downward propagation from the lower stratosphere to the near surface.*

For the NH, downward coupling in the multi-model mean is too strong at short and lags and to a lesser degree at longer lags (Figs. 4a, 13, 15), for both a regression approach and a composite approach based on extreme events. In contrast, a correlation approach indicates that biases are relatively small in the multi-model mean (Figs. 13, S2; consistent with Lee and Charlton-Perez, 2024). This difference in the overall conclusion as to whether downward coupling is biased among the different methodologies is likely due to variance in Zcap at 850 hPa that is too strong in most models (Fig. 2). Regardless of the methodology, downward coupling from the mid-stratosphere to the near surface is of reasonable strength in the multi-model mean. The multi-model mean coupling strength is the net effect of qualitatively different behaviors across models, however, and this metric is the most biased (in an absolute sense) of any across models. Downward coupling is stronger in models with poor climatological stationary waves, low tropospheric vertical resolution, or too long a persistence timescale of lower-stratospheric temperature anomalies (Figs. 6f and S9f). This sensitivity to climatological stationary waves is consistent with the known damping on annular-mode variations provided by planetary waves (Feldstein and Lee, 1998; Lorenz and Hartmann, 2003), though planetary waves may couple with vortex perturbations directly and act to bring vortex perturbations down to the surface (Song and Robinson, 2004; Simpson et al., 2013; White et al., 2020).

For the SH, downward coupling of polar-cap height from the lower stratosphere to the surface is too strong in most models (Figs. 4b, 13b, 15), even as polar-cap temperature anomalies decay too fast in these models (Figs. 4b, 14b). Hence too strong a downward coupling likely reflects overly strong eddy feedback in the SH (while the NH eddy feedback has an opposite signed bias, namely it is too weak), as has been recently shown for a subset of these models (Garfinkel et al., 2024).

The results above are based on relatively short hindcast periods so that the ERA5 correlations/regressions shown throughout may be subject to sampling variability. Indeed, Lawrence et al. (2023) showed that similar coupling metrics in GEFSv12 largely fell within the range of ERA5 sampling variability. Here we assume that since the S2S systems are initialized with the same internal variability as observed in the real world and are intended to be useful for predicting on subseasonal timescales, they should be able to reproduce the

ERA5 values (subsampled for each hindcast), and documenting the deviations from these values particularly in a multi-model comparison still enhances understanding of where and how the models are biased. Nonetheless, some of the model biases shown here fall within the range of ERA5 sampling variability (which can be estimated using the vertical black bars on, e.g., Figs. 12 and 13).

The NH polar vortex in these forecasting models is insufficiently coupled to tropospheric variability, consistent with too weak an impact of predictable tropospheric modes of variability such as the Madden–Julian Oscillation and snow cover anomalies on the vortex that has been documented in previous work using a subset of these models (Domeisen et al., 2020b; Garfinkel et al., 2020; Schwartz and Garfinkel, 2020; Stan et al., 2022). This conclusion is consistent with Lee et al. (2020), who also found that models systematically underestimate the stratospheric heat flux and vortex response to a Ural-blocking-like pattern. In contrast, the SH stratospheric vortex is realistically coupled with tropospheric variability. Interestingly, an older generation of chemistry-climate models analyzed by Eyring et al. (2006) displayed the correct stratospheric response of polar temperatures to wave forcing in the Northern Hemisphere but not in the Southern Hemisphere. However, their conclusions are based on 20 years of seasonal mean data in free-running atmospheric simulations without an attempt to rigorously quantify uncertainties. Here, we are focusing on shorter timescales and initialized forecasts and have orders of magnitude more data per model, which allow for a more stringent criteria of fidelity.

Downward coupling from 100 to 850 hPa is too strong in both hemispheres in the multi-model mean, though a few models have an opposite signed bias (e.g., NCEP, JMA CP3, and ISAC-CNR). While we link this in our study to biases in synoptic-eddy feedback, persistence of lower-stratospheric temperature anomalies, and quasi-stationary waves, there are other possible causes that might be relevant. Specifically, stratospheric ozone-circulation coupling is crucial in the SH spring and summer and also plays an important role in the NH spring. Some studies have shown that using prescribed ozone that includes year-to-year variations instead of climatological ozone improves the SH forecast skill of the surface climate (Hendon et al., 2020; Oh et al., 2022). Experiments with fully interactive ozone show further improvements in the representation of the tropospheric response (Romanowsky et al., 2019; Friedel et al., 2022a, b), although the downward coupling in models with interactive ozone is also strongly affected by model biases (Bergner et al., 2022). Future work should explore the role of prognostic or interactive ozone in S2S operational systems for downward coupling and improvements in predictive skill.

A poor representation of gravity waves is known to degrade stratosphere–troposphere coupling (Shaw and Perlwitz, 2010; Wicker et al., 2023); however the S2S archive does not include gravity wave drag as a standard output, and

even models with ostensibly similar parameterizations can nonetheless have very different net fluxes (Lott et al., 2024). Future work should evaluate the role of gravity waves for coupling strength should the requisite output be made available.

We find that the models better capture wave 2 vertical coupling from 500 to 100 hPa, likely because the biases in their climatological wave 2 heat flux are smaller than for wave 1. This appears to be contrary to climate models, which struggle more with wave 2 in the NH (it is typically too weak) and also tend to overestimate the number of wave 1 SSW events with respect to wave 2 events. Nevertheless, there is a notable bias in the coupling of wave 2 between 100 and 500 hPa at negative lags in the NH (second row of Fig. 3, lags −6 to 0). Specifically, strong values of 100 hPa heat flux have a weak tendency to precede pulses at 500 hPa; however only one model captures this effect. This may reflect problems more generally with downward wave coupling and/or wave reflection; exploring this possibility in greater detail is left for future work.

We have formulated a reduced set of key metrics and diagnostics that can be saved and analyzed relatively easily as part of the model development cycle to serve as a benchmark. We hope this set of diagnostics will be adopted and will aid the development of improved models. We also want to emphasize that this analysis is only possible with the output of stratospheric data. The relative paucity of levels makes it difficult to more fully diagnose why the upward-coupling strength and downward-coupling strength within the stratosphere is too weak in most models. For example, this bias could be related to biases in the representation of the tropopause and lowermost stratosphere (Weinberger et al., 2022); however such an effect is impossible to diagnose with data only at 200, 100, and 50 hPa. Finally, the implications of poor coupling for surface climate and predictability in specific regions where the stratosphere is known to have a large impact need to be explored.

*Data availability.* The hindcasts from the S2S database used here are available from https://apps.ecmwf.int/datasets/data/s2s/ TS5 under the "Reforecasts" S2S set. The NOAA GEFSv12 hindcasts can be obtained from https://registry.opendata.aws/noaa-gefs-reforecast/ TS6. Hindcasts for CESM2–CAM are available at https://www.earthsystemgrid.org/dataset/ucar.cgd.cesm2.s2s_hindcasts.html TS7, while those for CESM2–WACCM are from https://www.earthsystemgrid.org/dataset/ucar.cgd.cesm2-waccm.s2s_hindcasts.html TS8.

*Supplement.* The supplement related to this article is available online at: https://doi.org/10.5194/wcd-7-1-2025-supplement.

*Author contributions.* CIG and AHB drafted the paper. CIG produced the final version of all figures except Figs. 10 and 11. ZDL

organized and led the SNAP effort leading to this paper and also downloaded all of the data. AHB produced the final version of Figs. 10 and 11. EDS produced an earlier version of Figs. 10 and 11. IS and AYK produced an earlier version of Fig. 13. GK produced earlier versions of Figs. 5 and 6. All the listed co-authors were active participants in this SNAP community effort and provided comments on the draft manuscript.

*Competing interests.* At least one of the (co-)authors is a member of the editorial board of *Weather and Climate Dynamics*. The peer-review process was guided by an independent editor, and the authors also have no other competing interests to declare.

*Acknowledgements.* This work uses S2S project data. S2S is a joint initiative of the World Weather Research Programme (WWRP) and the World Climate Research Programme (WCRP). This work was initiated by the Stratospheric Network for the Assessment of Predictability (SNAP), a joint program of APARC (WCRP) and the S2S project (WWRP–WCRP).

Chaim I. Garfinkel and Jian Rao are supported by the ISF–NSFC joint research program (Israel Science Foundation grant no. 3065/23 and National Natural Science Foundation of China grant no. 42361144843). Chaim I. Garfinkel and Judah Cohen are supported by the NSF–BSF joint research program (National Science Foundation grant no. AGS-2140909 and United States–Israel Binational Science Foundation grant no. 2021714). Irina Statnaia and Alexey Y. Karpechko are supported by the Research Council of Finland (grant no. 355792). The work of Marisol Osman is supported by UBACyT (project nos. 20020220100075BA, PIP 11220200102038CO, and PICT-2021-GRF-TI-00498 TS9). The work of Alvaro de la Cámara is funded by the Spanish Ministry of Science, Innovation and Universities (project no. PID2022-136316NB-I00). Marta Abalos, Blanca Ayarzagüena, and Natalia Calvo acknowledge the support of the Spanish Ministry of Science, Innovation and Universities through the RecO3very project (no. PID2021-124772OB-I00). Froila M. Palmeiro and Javier García-Serrano have been partially supported by the Spanish AT-LANTE project (no. PID2019-110234RB-C21) and Ramón y Cajal program (no. RYC-2016-21181), respectively. Neil P. Hindley and Corwin J. Wright are supported by the UK Natural Environment Research Council (NERC; grant no. NE/S00985X/1). Corwin J. Wright is also supported by a Royal Society University Research Fellowship (grant no. URF/R/221023). Seok-Woo Son and Hera Kim are supported by the National Research Foundation of Korea (NRF) grant funded by the Korean government (Ministry of Science and Information and Communication Technology, MSIT) (grant no. 2023R1A2C3005607). The work of Rachel W.-Y. Wu is funded through ETH (grant no. ETH-05 19-1). Daniela I. V.

Domeisen gratefully acknowledges support from the Swiss National Science Foundation (project no. PP00P2_198896).

This material is based upon work supported by the U.S. Department of Energy Office of Science Biological and Environmental Research (BER) program Regional and Global Model Analysis (RGMA) component of the Earth and Environment Systems Modeling program (award no. DE-SC0022070) and National Science Foundation (NSF; grant no. IA 1947282). This work was also supported by the National Center for Atmospheric Research (NCAR), which is a major facility sponsored by the NSF (cooperative agreement no. 1852977). Zachary D. Lawrence was partially supported by NOAA (award no. NA20NWS4680051).

*Financial support.* This research has been supported by the Israel Science Foundation (grant no. 3065/23), the United States–Israel Binational Science Foundation (grant no. 2021714), and the National Science Foundation (grant no. AGS-2140909). TS10

*Review statement.* This paper was edited by Thomas Birner and reviewed by Edwin Gerber and one anonymous referee.

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

**Remarks from the language copy-editor**

**Remarks from the typesetter**