# Peer review of "A process-based evaluation of biases in extratropical stratosphere-troposphere coupling in subseasonal forecast systems"

_EGUsphere, 2024_

## Author Comment (AC1)

**Reviewer 1**

This paper discusses the ability of long range forecast models to represent troposphere stratosphere coupling. The results come from a set of state of the art prediction systems and present an interesting set of results and metrics that could be used by operational centres developing future systems. I have most minor comments and suggestions.

We thank the reviewer for their constructive comments.

L30: The celebrated criterion of no wave propagation above an upper wind threshold (Charney and Drazin 1961) is a linear rather than nonlinear result.

**We have added a citation to the end of the previous sentence about nonlinearity (Boljka and Birner 2020) to make it clearer that the sentence about Charney and Drazin is not a continuation of the previous idea.**

L64-66: Suggest you remove this summary of results as it is repetitive of the Abstract and pre-empts the results section.

**We have shortened this sentence.**

L128: Why does an error in variance affect the correlations? After all, correlations are by definition insensitive to the amplitude of variability so do you mean regressions here? ;,

**A model with a too-small coupling regression coefficient but a too weak variance bias that is even more dramatic, can have a too-strong correlation bias. We give several examples of this in the Results section (e.g., upward coupling for BoM). We have added "(the Results section provides several examples of such behavior)".**

Fig.1 is very striking and suggests very large errors in the total variance of the models. Is it really correct that there are tens of percent errors in variance with too much in the troposphere and too little in the stratosphere? I have not seen this before and I think you should check and then emphasize this if it's robust.

**We have computed the bias in the variance of polar cap geopotential height (Zcap) for additional tropospheric levels, and have confirmed that the models are systematically biased high at all levels up to 300hPa. We have confirmed this by creating histograms of Zcap for a few select models; the PDF of Zcap is indeed wider in the models than in ERA5. See below for Zcap500 for IFS.**

[Figure]

**Between 300hPa and 100hPa the models have a mix of biases, and then above 100hPa the variance is systematically too low. (This is compared to ERA5 subsampled to each model's available dates). We are not aware of any paper documenting the too-strong variance bias in the troposphere, and have now added mention of this into the methods section "We are not aware of previous work that has found such too-strong variance biases in the troposphere, and the causes and implications of these biases should be explored in future work.". (That models suffer from too-weak variance in the stratosphere is better known, and is worse in low-top models.)**

Fig.2 would benefit from adding N Hem and S Hem labels.

**The latitude range is indicated in the figure title**

L155 and throughout the paper: In many cases it is really only some of the models that show the errors highlighted, for example in Fig.2a. PLease can the paper be phrased more carefully to say things like "models in general" or "models tend to" to avoid giving the impression that all models show the same errors?

**We now use "most", "generally", etc.**

L185-190, L245 etc: The paper tends to only reference very recent papers rather than giving a representative picture of current knowledge and following the *scientific convention of acknowledging those papers that first demonstrated ideas*. Some rewriting

is needed to better represent this. For example some wider discussion on the current knowledge of the effects of model lid height/degraded stratosphere would be welcome to put the results in wider context. Papers by Boville, J.A.S., 1984; Lawrence, J.G.R.,1997; Marshall and Scaife, J.G.R., 2010; Shaw and Perlwitz J.Clim., 2010.

**These papers are now cited, though we prefer to include them in the introduction rather than in the results.**

L265, L400: The underestimation of the heat flux variability in the stratosphere and upper troposphere is interesting. Is the underestimation of v*T* related to the underestimation of ENSO teleconnections reported in Garfinkel et al 2022 and Williams et al 2023? IS this also related to the so called signal to noise paradox in long range forecasts which appears to be clearer in the northern hemisphere than the southern hemisphere, just like the biases reported here? Perhaps some discussion would be useful on these points?

**We indeed think that the lack of heat flux extremes are related to a poor simulation of teleconnection processes (and potentially related to ENSO), and we include suggestions to this effect near line 271. We have added to the discussion section near line 405 "(e.g., one possibility might be poor simulation of teleconnections arising from ENSO; Garfinkel et al 2022, Williams et al 2023)"**

**Garfinkel et al, in press finds no evidence for a signal to noise paradox in the stratosphere in the 7 S2S models they examined (7 of the 22 examined here). We are currently writing a follow-on paper which will examine the signal to noise paradox in all 22 of these model versions, but preliminary work indicates no S2N paradox in the stratosphere in any of these models. This follow-on work will include a much more detailed discussion of signal to noise characteristics of these models.**

Figure 10: please provide a full caption for ease of reading.

**We have provided a full caption (this is now Figure 11 in the revised version).**

L362-364: Please again provide wider referencing for the surface impact of the stratosphere e.g. Baldwin and Thompson Quart. J Roy. Met. Soc. 2009, Kidston et al., Nat. Geosci., 2015.

**added**

L450: This is a potentially important point and should be moved to the earlier methods section.

**We have added this to the methods section as well (near line 89)**

---

## Author Comment (AC3)

**Reviewer 2**

The authors present a tour-de-force evaluation of stratosphere-troposphere coupling in subseasonal forecast systems. As I understand, this comprehensive analysis is the product of a team effort, led by Chaim Garfinkel with Zach Laurence and Amy Butler, as part of the SNAP project. I recommend publication of this manuscript after consideration of the minor points below. This study will provide a valuable reference point for evaluating subseasonal to seasonal (S2S) forecasting systems, both for assessing the current state of systems, but more importantly, for identifying key metrics that modeling centers can use to measure improvement in the future. Furthermore, I found the manuscript well written and structured. The amount of information in the figures was at times overwhelming, but I appreciate that the goal is to document the state of the S2S systems. I commend the authors for both the thoroughness and quality of the analyses and presentation.

**We thank Dr. Gerber for his constructive comments.**

Minor points to consider

1) I trust that the lid height is not the most important feature of an S2S system. Rather, as the authors are fully aware, it is correlated with other features that matter. (This point was made clear when they had to decide what to do with WACCM, which has a much higher top than the other models. And to take it to the extreme, I trust that adding a layer in the thermosphere to any low top model will have little, if any, impact on that model's performance in the stratosphere.) I trust that models with a higher lid have better resolution in the stratosphere, better representation of sub-grid physics relevant to the stratosphere (gravity wave drag), and perhaps most importantly, indicate an interest in stratospheric dynamics from the relevant modeling center, so that care was taken to capture and evaluate performance of the model in this region.

The paper is already long, but some discussion, and possibly a little analysis, might help make this point clear. To be concrete, I would suspect that the resolution in the UTLS (upper troposphere and lower stratosphere), and more generally through the stratosphere, is most critical. A high lid height's main contribution is likely to ensure that the needed numerical sponge layers at the top are above the stratosphere, where they would corrupt the dynamics.

To be constructive, could the authors provide a bit more information in Table 1? For example, what is the vertical resolution near the tropopause, and how many layers are included between 100 and 10 hPa, and 10 and 1 hPa.

**We have added the requested information to the new Figure 1 (copied below)**

[Figure]

**Figure 1.** Schematic representation of model vertical resolution for all S2S prediction systems used in this study. Each block represents the pressure range indicated on the y axis. The number of model levels in each range is shown numerically. The red number at the top of each bar shows the total number of levels in each model.

Emphasizing that my suggestions are minor, it could also be important to identify features in the models, e.g., their representation of gravity wave momentum parameterizations (types - orographic, frontal, convective) and how well its tied to sources (fixed sources, or dynamic, e.g., coupled with convection/frontegenisis). Might it also be possible to note how radiation is treated (particularly, ozone).

**Trying to meaningfully compare different gravity wave parameterizations across models is a difficult task, as two models with ostensibly very similar schemes can have very different net fluxes (e.g. Lott et al 2024 in the context of non-orographic GW for half the number of models that we include). We are hesitant to attempt a similar analysis here. We have added to the discussion near line 486: "Finally, a poor representation of gravity waves is known to degrade stratosphere-troposphere coupling (Shaw and Perlwitz 2010), however the S2S archive does not include gravity wave drag as a standard output and even models with ostensibly similar parameterizations can nonetheless have very different net fluxes (Lott et al 2024). Future work should evaluate the role of gravity waves for coupling strength should the requisite output be made available."**

**Essentially all S2S models still use climatological ozone (NCEP is a notable exception). We note this in the discussion section. A future SNAP activity might be devoted to the importance of ozone for forecast skill, but we prefer not to elaborate for now.**

Lott, François, Raj Rani, Charles McLandress, Aurélien Podglajen, Andrew Bushell, Martina Bramberger, H-K. Lee et al. "Comparison between non orographic gravity wave parameterizations used in QBOi models and Strateole 2 constant level balloons." *Quarterly Journal of the Royal Meteorological Society* (2024).

Finally, I'm curious if vertical resolution (grid spacing) or the number of levels in the stratosphere might be better metrics for comparison than lid height, say, in Figure 4. I

appreciate this could be a rabbit hole – best left for future research, if at all – but more quickly, one could plot lid height vs resolution or number of levels, to quickly get a sense of how these things are correlated.

**We have added three figures to the supplemental material that perform the requested analysis. Namely, we contrast coupling strength with the number of levels below 300hPa, between 300hPa and 100hPa, and between 100hPa and 10hPa. The one notable effect is that better resolution in the tropopause region is associated with better relaxation timescales for T100. This makes sense if these relaxation timescales are dictated by poor representation of transport etc. For the other metrics and levels we don't see much evidence for a particularly stronger effect than we found when we just considered the model lid and v'T'500hPa climatological biases. More generally, the number of model levels between 100 and 10hPa isn't very well correlated to biases in the coupling processes, but the number of levels below 300hPa (and to some extent between 100 and 300hPa) is much more so. Maybe this suggests that better simulating/resolving tropospheric processes is really key to reducing some of the biases in the coupling that we are seeing, rather than moving to much higher vertical resolution in the stratosphere. The revised paper will discuss this in more detail.**

[Figure]

**Figure S5.** As in figure 4 in the main text but for number of model levels below 300hpa.

[Figure]

**Figure S6.** As in figure 4 in the main text but for number of model levels between 100 and 300hpa.

[Figure]

**Figure S7.** As in figure 4 in the main text but for number of model levels between 10 and 100hpa.

To end on a positive note, I appreciated how the authors explain that many of the correlations with lid height are even stronger if you correlated with the mean state (i.e., wave 1 variability differences correlate strongly with biases in the mean representation

of wave 1).  This provides a concrete pathway for trying to improve models, other than simply raising the lid!

2) It was at times hard for me to assess the sampling uncertainty in results.  I put this as a minor point, as the goal here was to document differences between models, not to say that one model was "better" than another, or clearly wrong.  Below are some comments meant to be helpful, not to nit pick.

The winter stratosphere is one of the most variable regions of the atmosphere, and work on vortex variability has been hampered by sampling uncertainty.  From figure 1 onwards, I was unsure how much of a bias indicates a problem, as opposed to just bad luck.  At the S2S time scale, we expect the atmosphere to be entering a chaotic regime, so I don't think it's fair to say that models must reproduce ERA5.  Rather, they should be able to reproduce the statistics of ERA5.

To be constructive, you have a lot more data in ERA5.  Might it be possible to evaluate metrics over 1979-1999, to give a very rough estimate of how much the "truth" changes depending on the period?   If these two periods in ERA5 differ by X %, that would be a very rough estimate of sampling uncertainty.

In many of the figures, e.g. 2, 7, 11, the authors show how the answer in ERA5 changes when data is subsampled as with the forecast systems.  This is exactly the type of analysis I would like to see, but I didn't fully understand how they did it.  Maybe just a paragraph in the methods section would help.  Also, in the text,, I didn't see too much discussion of these sampling error bars.  For instance, in the discussion surrounding Fig 2a and b, at line 156 the authors state that the models systematically underestimate the correlation and regression coefficient of wave 1.  I agree that most models fall below the thick black line, but don't many fall within the sampling uncertainty here?  [As I emphasized above, this is a minor concern, as the authors are not trying to explicitly say that models are wrong, but rather establish a metric for comparision.]

**Thank you for bringing up this point.  The black vertical bar in these figures does indeed indicate the sampling uncertainty in ERA5, and we have added this to the text in several locations (e.g., in the methods "Nonetheless, these thin vertical lines offer an estimate of the range of sampling variability in  ERA5, and thus if a given model lies outside of this range, a bias can be even more confidently detected."). However, for the pixel plots, we compare to ERA5 subsampled to each model, as ultimately S2S models are intended not only to capture the statistics of ERA5, but also should be useful for forecasting purposes of specific events. That is, these models are initialized with the same internal variability as observed (unlike climate models) and are intended to be useful for operational forecasting, hence it is reasonable to hope that the models can capture the coupling strength evident in ERA5 over the identical dates (in contrast to climate models when there isn't a clear correspondence to the observed atmospheric evolution over specific dates).**

**We have clarified how these thin vertical lines are computed: "For figures showing lagged correlations and lagged regression, we show the mean across the forecasting systems of the subsampled coupling strength with a solid black line, and the spread in the subsampled coupling strength across the available S2S systems with a vertical thin line"**

**In our discussion of Figure 2 (now Figure 3) specifically, we now write "While the forecast systems capture this behavior qualitatively, most underestimate the magnitude of the correlation and regression for wave-1"**

Below are small questions about statistics.

1. At line 170, you talk about a correlation of -0.34. The caption notes that a correlation of -.42 is needed to reject the null hypothesis at 95% confidence. I suspect there's a real problem with the models, but it would be good to acknowledge that this could be by chance in the text.

**We now note that this correlation of -0.34 is not significant.**

2. If you consider just the high top models, is there any significant correlation with lid height? If not, then it's strong evidence that once the lid is sufficiently high to get the sponge out of the stratosphere, lid height doesn't matter any more.

**A version of Figure 5 but with the correlation calculated after excluding all low top models is below. The correlation is in blue if it differs in absolute value from the correlation using all models by 0.05. For downward coupling, none of the correlations are statistically significant. For upward coupling there is still a tendency for higher high-top models to better represent upward wave-2 (panels b, h). For the sensitivity of the vortex to upward v'T' (panel c), higher high-top models perform worse, however this is almost entirely due to one outlier model (WACCM).**

**We have added to the discussion section "We also re-calculated the metrics in Figure 5 but for high top models only, and found that there is still a tendency for higher high-top models to better represent upward wave-2 (not shown). However, for other metrics the correlations are reduced or even change sign, suggesting that once lid is sufficiently high, the effect of lid height becomes saturated."**

[Figure]

3. For Figures 6 and 8, I wonder if the fact that models provide ensemble forecast could weaken the correlation. [As I understand, at least for figure 8, the correlation was first computed for each ensemble member, and then averaged; I'm not sure if something similar was done for 6.] Wouldn't this have a tendency to reduce the

correlation for the models: you are comparing a model mean against a single sample from ERA5.  To be constructive, do the magnitudes of the correlation from the models increase to similar values if you consider only one ensemble member?  Or could you put a rough uncertainty estimate on the ERA5 values?

**Indeed, the regression coefficients for figure 6 are computed for each ensemble member, and then averaged. We have now added this.**

**While it is true that averaging over the correlation coefficients might tend to dampen the overall correlation from a randomly chosen member (unless one performed a fisher z transformation first), we are showing regression coefficients in these figures. Namely, we compute the regression coefficient from each member and then average them. This shouldn't lead to an underestimate, and won't suffer from the same issue that correlations suffer from in that correlations cannot exceed 1 (barring a fisher-z transformation). We think that such an average over regression coefficients is a better estimate of a given model's coupling strength than simply picking a member at random.**

4. Line 215 and Figure 7.  As I noted about the sampling error bars on figure 2, they don't seem to factor into the discussion of figure 7 here.  I agree the models are below the thick black line, but many seem within the sampling uncertainty.

**In the NH, essentially all models lie below the vertical black lines for the regression coefficients, so we use the language "Most models underestimate …." The pixel plots indicate that this bias is pervasive if we subsample ERA5 to match the models.**

To end on a positive note, Figure 9 was particularly eye-catching.  I was initially worried about the statistics of extreme events, but this figure makes a pretty compelling argument that something is wrong with the models.  It begs future work on the spread of the S2S systems with time: they are both drifting to a biased mean state, but seem to be losing a lot of variability!

**Thank you for the positive comments!**

—

Tiny comments

I believe that "e.g." should always be followed by a comma, e.g., as I just demonstrated.  A quick search would spot ones you missed.

**corrected**

At line 98, a closing parentheses is missing, or you could just eliminate the first one before e.g.

**corrected**

311 and 439:  I did not find this long list of citations about how we don't  understand planetary waves very informative.  Consider being a bit more sparse here, or breaking up the list, explaining briefly the key contributions of the citations.

**The lists have been shortened**

I felt the opening line of the discussion and conclusions could be flipped.  I'd first note that the troposphere perturbs the stratospheric circulation (i.e., there's a reason why stratospheric variability differs so much between the hemispheres), and then that stratosphere in turn impacts the troposphere.

**We assume you are referring to the second sentence in the discussion. We have made the requested change.**

At line 425 I thought that recent work by Marina Friedel and Gabriel Chiodo was particularly relevant.  The authors mention this work shortly afterwards, at line 470, so I think it's fine.

**We considered citing it here too, but decided that it more clearly belongs near line 470.**

—

To finish my review, I wanted to note that the authors very nicely sum up the importance of this work in the last paragraph.  Bravo!

**Thank you!**

---

## Author Response (AR2)

- line 29/30 (track-changes version): would be good to add Scinocca & Haynes (1998), who I think were the first to suggest this possibility (Scinocca, J. F. and Haynes, P. H.: Dynamical Forcing of Stratospheric Planetary Waves by Tropospheric Baroclinic Eddies, J. Atmos. Sci., 55, 2361–2392, 1998.)

**We have added the requested reference**

- following up on one of the comments by reviewer 2 about the way the model correlation or regression coefficients are computed: I think it would help to clarify this procedure early in the manuscript, perhaps in the methods section (i.e., mention that you compute relevant statistic for each ensemble member first, then average over members - unless I missed something, the order of computation certainly doesn't matter for the variance or covariance, but it does for the regression and correlation coefficients)

**We have added such a sentence near line 135**

- new Fig. 1 is very helpful and I like the way you've neatly summarized the relevant information; for the lighter colors I find it hard to read the white text embedded within them - perhaps use black font for those (and clarify in caption that font color has no additional meaning)?

**Changed as suggested**